# FESOM2.1-REcoM3-MEDUSA2: an ocean-sea ice-biogeochemistry model coupled to a sediment model

Ying Ye[1], Guy Munhoven[2], Peter Köhler[1], Martin Butzin[1,3], Judith Hauck[1], Özgür Gürses[1], and Christoph Völker[1]

[1]Alfred-Wegener-Institut Helmholtz-Zentrum für Polar- und Meeresforschung (AWI), P.O. Box 12 01 61, 27515 Bremerhaven, Germany
[2]Laboratoire de Physique Atmosphérique et Planétaire, Université de Liège, B–4000 Liège, Belgium
[3]MARUM — Center for Marine Environmental Sciences, University of Bremen, Bremen, Germany

**Correspondence:** Ying Ye (Ying.Ye@awi.de)

**Abstract.**

This study describes the coupling of the process-based Model of Early Diagenesis in the Upper Sediment (MEDUSA version 2) to an existing ocean biogeochemistry model consisting of the Finite-volumE Sea ice-Ocean Model (FESOM version 2.1) and the Regulated Ecosystem Model (REcoM version 3). Atmospheric $CO_2$ in the model is a prognostic variable which is determined by the carbonate chemistry in the surface ocean. The model setup and its application to a pre-industrial control climate state is described in detail. In the coupled model, 1390 $PgC$ are stored in the top 10 $cm$ of the bioturbated sediment, mainly as calcite, but also to 10% as organic matter. Simulated atmospheric $CO_2$ stabilizes at $\sim 295$ $ppm$ after 2000 $years$ of the coupled simulation, in line with the $CO_2$ level expected from the climate forcing conditions. Sediment burial of carbon, alkalinity and nutrients in the coupled simulation is set to be compensated by riverine input. The spatial distribution of biological production is altered depending on the location of riverine input and reduction of sedimentary input as well as the strength of local nutrient limitation, while the global productivity is not affected substantially. With this coupled ocean–sediment system the model is able to simulate the carbonate compensation feedback under moderate perturbation of $CO_2$ in the atmosphere.

## 1 Introduction

The ocean plays a key role in the global carbon cycle. It stores about 37,200 PgC (Keppler et al., 2020), more than 40 times as much carbon as the atmosphere, which contained 884 PgC (or 417 ppm) in the year 2022 (Lan et al., 2023). About 25–30% of the global anthropogenic $CO_2$ emissions are taken up by the world oceans (Friedlingstein et al., 2022).

$CO_2$ enters the ocean through gas-exchange, where it dissolves in seawater. A unique feature of dissolved $CO_2$ is that it reacts with water to form carbonic acid ($H_2CO_3$), which is instable and dissociates as a function of temperature, salinity and pressure into bicarbonate ($HCO_3^-$), carbonate ($CO_3^{2-}$) and hydrogen ($H^+$) ions (Zeebe and Wolf-Gladrow, 2001). The dissolved inorganic carbon (DIC), which is the sum of $CO_2$, $HCO_3^-$ and $CO_3^{2-}$, is distributed in the ocean via circulation. Part of the carbon in the surface ocean is also taken up via photosynthesis by marine phytoplankton and exported into the ocean interior via the sinking of dead organic matter. When stored in the deep ocean, this carbon reduces the surface concentration of

DIC and allows for further $CO_2$ uptake from the atmosphere. Another important process in the marine carbon cycle is driven by calcifying plankton. These organisms produce calcium carbonate ($CaCO_3$) shells whereby $CO_2$ is released back into the

atmosphere. These processes which all influence the surface-to-depth-gradient in DIC are also summarized as the so-called marine carbon pumps (Volk and Hoffert, 1985). Some of the particulate carbon (i.e., particulate organic carbon and $CaCO_3$, ca. 1% of primary production; Sarmiento and Gruber, 2006) escapes dissolution and remineralization in the water column and sinks to the seafloor, where it might be buried. These particles are then removed from the relatively fast cycling of carbon at the surface of the Earth.

The storage of carbon, alkalinity and nutrients in sediments introduces an additional slow timescale to carbon cycling, and overall increases the carbon storage in the sediment-ocean system. The global burial flux of particulate organic carbon (POC) in marine sediments has been reported to be in a range of $160$–$2600\,\mathrm{PgC\,kyr^{-1}}$ (Burdige, 2007; Dunne et al., 2007; Sarmiento and Gruber, 2006; Muller-Karger et al., 2005). In total $\sim 280\,\mathrm{PgC\,kyr^{-1}}$ are buried as $CaCO_3$ in marine sediments of which $100$–$150\,\mathrm{PgC\,kyr^{-1}}$ find their way into sediments of the deep-sea below at least $1\,\mathrm{km}$ of water depth (Sarmiento and Gruber,

2006; Dunne et al., 2012; Cartapanis et al., 2018; Hayes et al., 2021). Furthermore, marine sediments play an important role as they provide records of the Earth's past climate. They react via the carbonate compensation feedback to any changes in the marine carbon cycle, in which the deep ocean carbonate ion concentration is brought back to its initial values after a perturbation on a multi-millennial timescale via sediment dissolution of $CaCO_3$ (Broecker and Peng, 1987).

Anthropogenic carbon emissions represent a rapid carbon cycle perturbation and in high-emission scenarios (Meinshausen

et al., 2011), may ultimately lead to the massive dissolution of $CaCO_3$ in seafloor sediments over the next millennia (Archer et al., 1997). This carbonate compensation feedback contributes to a reduction of the long-term airborne fraction of anthropogenically emitted $CO_2$ from more than 20% if only the atmosphere-ocean is considered to be less than 10% (Archer et al., 2009; Köhler, 2020). This additional oceanic uptake of anthropogenic carbon through the dissolution of $CaCO_3$, however, operates on a multi-millennial timescale, and is therefore only of interest for the geological fate of fossil emissions, but not for our

near future. Hence, understanding processes controlling the sediment-ocean exchange and quantifying the carbon storage in marine sediments are crucial to explain transient behaviour over changing climates, e.g. the glacial-interglacial $CO_2$ variations (e.g. Brovkin et al., 2012; Köhler and Munhoven, 2020), and to predict the long-term ocean sequestration of anthropogenic carbon (Archer et al., 2009; Köhler, 2020).

All ocean biogeochemistry models incorporate a scheme to describe the fate of biogenic material that reaches the seafloor,

but differ in their complexity (Munhoven, 2021, and references therein). The most simple schemes start from a reflective boundary condition, where all material reaching the seafloor is remineralized and returned to solution. More complex schemes consider a single, vertically integrated mixed-layer sediment box with a complete mass balances for the particles settling to the seafloor. Even higher complexity is found in vertically resolved sediment models describing diagenetic reactions, mechanical changes of dissolved and solid components as well as burial fluxes out of the surface sediment.

FESOM2.1-REcoM3, consisting of the Finite-volumE Sea ice-Ocean Model 2.1 and the Regulated Ecosystem Model 3, is one of the ocean biogeochemistry models, which so far includes a simple one-layer sediment model (Gürses et al., 2023). REcoM3 describes the marine ecosystem at medium complexity with two phytoplankton classes including silicifiers and calcifiers,

two zooplankton classes representing mixed zooplankton and polar macrozooplankton, and considers flexible stoichiometry of C, N, Si, Fe, $CaCO_3$, and chlorophyll. Various iron sources (sediment, dust and rivers) are implemented into REcoM3 and the model also has the option to simulate the cycles of $^{13}C$ and $^{14}C$ (Butzin et al., 2023). The sediment box used so far in REcoM3 ensures the mass conservation by a complete remineralization of material sinking into the box. It represents processes in the surface sediment and is useful for short-term simulations, since the characteristic time scales of early diagenetic processes are often of the order of days to months, while long-term burial via sedimentation (which compensates riverine inputs from continental weathering) acts on time scales of thousands of years. Kriest and Oschlies (2013) have shown that the introduction of a sediment box makes models more robust against the uncertainties of the remineralization length scale, compared to models that remineralize everything in the water column. However, without considering sediment-ocean fluxes and feedbacks in more detail the model would not be able to reasonably simulate transient changes over glacial/interglacial timescales.

In Sect. 2), we describe the coupling of FESOM2.1-REcoM3p, a model configuration targeted for paleo-application, with MEDUSA2, the Model of Early Diagenesis in the Upper Sediment (Munhoven, 2021). MEDUSA2 is a process-based sediment module that offers a complex alternative to the previously used simple one-layer sediment. This is the first realisation of such an ocean-sediment setup of the marine carbon cycle with flexible stoichiometry of organic matter. In comparable existing alternatives (e.g. Kurahashi-Nakamura et al., 2020; Moreira Martinez et al., 2016) stochiometry was kept fixed. This feature enables our model to simulate the growth limitation and community composition of phytoplankton in a more realistic way so that the biological carbon pump has a higher flexibility to react to climate change (Seifert et al., 2022; Schartau et al., 2007; Hohn, 2009). The final model configuration, referred to as FESOM2.1-REcoM3p-MEDUSA2, is applicable to relevant questions in paleoclimate research and should be able to provide new insights into the long-term dynamics of the marine carbon cycle. The coupled ocean-sediment simulation of this configuration under pre-industrial climate conditions is analysed in Sect. 3), along with transient simulations with perturbations in atmospheric $CO_2$, while its applications to question of the last glacial cycle are envisaged in future, more targeted studies.

## 2 Methods

### 2.1 Model description

#### 2.1.1 REcoM3p – A REcoM3 configuration for paleo research

REcoM is an ocean biogeochemistry and ecosystem model describing cycles of carbon, oxygen and nutrients (nitrogen, silicon and iron) with varying intracellular stoichiometry in phytoplankton, zooplankton and detritus. REcoM3 is the most recent release version and a detailed description of this version including its coupling to FESOM2.1 is given by Gürses et al. (2023). The configuration REcoM3p used here has on the one hand reduced complexity with respect to functional groups of the modelled ecosystem and considered only one generic zooplankton and one detritus class, instead of two in the full version of REcoM3 (Fig. 1). As in the full version, diatoms and small phytoplankton which include calcifiers (only calcite producers, no aragonite) are considered here. The total 22 biogeochemical tracers cover nutrients (DIN, DSi and DFe), two types of

phytoplankton (diatoms and small phytoplankton) with the state variables C, N and chlorophyll, as well as biogenic silica in diatoms and calcite in small phytoplankton, one zooplankton with C and N pools, one detritus with the state variables C, N, calcite and opal, dissolved organic matter with C and N pools, DIC, Alk and oxygen. The biological cycling of iron is described using a fixed Fe:N ratio in phytoplankton, zooplankton and detritus. The same parameter values were used as described in Gürses et al. (2023) and only two parameters were tuned for the reduced food web and coarser resolution (see Tab. D1 in the appendix).

So far REcoM3 only included a single-layer sediment. Particles sinking out of the bottom water boxes enter this sediment layer and go through remineralization (organic particles) and dissolution (calcite and opal) following a simple first-order decay approach: organic matter remineralization is neither dependent on $O_2$ availability nor does it follow different redox pathways; carbonate dissolution proceeds irrespective of the ambient saturation state (similarly to the dissolution in the water column). The approach is thus equivalent to a classical reflective boundary with temporal buffering. The fluxes of solutes back to the bottom water boxes are derived from the remineralization and dissolution rates of the solids via the elemental ratios that characterize them. While the main aim of this study is the replacement of this simple sediment with the more complex sediment representation of the MEDUSA2 model, we keep this configuration as an alternative option for comparisons (labelled $R_{sedbox}$ – see Sect. 2.4 below).

On the other hand, REcoM3p contains some extensions of relevance for the planned paleo applications when compared to the version described by Gürses et al. (2023): First, atmospheric $CO_2$ concentrations are calculated assuming that the atmosphere can be represented as a homogeneous carbon reservoir. The carbon cycle on land (continental biosphere) is not considered. Temporal changes in the atmospheric volume mixing ratio of $CO_2$ ($X_{CO_2}$, in ppm) then solely result from the globally integrated air-sea $CO_2$ flux, given by

$$\frac{\delta X_{CO_2}}{\delta t} = -\frac{\rho_{air}}{m_{textatm}} \cdot 10^6 \int F_{CO_2} \, dA,$$

where $F_{CO_2}$ ($\mathrm{mol\,m^{-2}\,s^{-1}}$) is the regional air-sea $CO_2$ flux (calculated according to Wanninkhof (2014)), $dA$ integrates over the ocean area, $\rho_{air} = 0.02897\,\mathrm{kg\,mol^{-1}}$ is the molar mass of dry air (from Picard et al. (2008), rounded here to four significant figures) and $m_{atm} = 5.1352 \cdot 10^{18}\,\mathrm{kg}$ is the mass of the dry atmosphere (Trenberth and Smith, 2005). The factor $10^6$ serves to convert from mol fractions to ppm.

Second, a riverine source of dissolved iron (DFe) was added to the already existing two sources from dust and marine sediments. Furthermore, due to the coupling to MEDUSA2, the sedimentary source of iron can be calculated in two ways: 1) in a fixed ratio to degradation of particulate organic nitrogen (PON) in the benthic layer as described in Gürses et al. (2023, Eq. A67 in Appendix A); 2) in a fixed ratio to the diffusive flux of dissolved inorganic nitrogen (DIN) calculated by MEDUSA2 in the coupled simulations. Elrod et al. (2004) demonstrated a clear correlation between the iron flux out of sediments and the oxidation of organic matter on shelves, with a $\mathrm{Fe:N}$ ratio that is much higher than typical $\mathrm{Fe:N}$ ratios in sinking organic matter. Under anoxic conditions in sediments, the flux of iron is increased due to the greater solubility of ferrous iron. To represent this effect, we applied a higher $\mathrm{Fe:N}$ ratio (3 µmol Fe : 20 mmol N) for the flux of iron from the sediment to the water column than the ratio of 1 µmol Fe : 30 mmol N that we used for remineralization in the water column. The same Fe : N

ratio is used for both methods to calculate the sedimentary input of iron. A comparison of source strengths for iron is discussed in Sect. 3.2.

Third, carbon isotopes were recently implemented into REcoM3p, as described in Butzin et al. (2023). However, the implementation of carbon isotopes into MEDUSA2 is not yet finished, which is why we here use REcoM3p with carbon isotopes switched off.

When coupling REcoM3 to FESOM2.1 there remain some minor tracer conservation issues, that are related to the use of an unstructured grid and need to be addressed. Although small (e.g., $0.53\%\mathrm{kyr}^{-1}$ for the global Si inventory in the ocean, i.e., $0.48\,\mathrm{TmolSi\,yr}^{-1}$, which is smaller than the uncertainty on most input and output fluxes to and from the ocean (Tréguer et al., 2021)), such imbalances may accumulate in an unfavourable way during simulation experiments run for tens of thousands of years and longer. We therefore included a spatially uniform mass correction at the end of each coupling cycle (every 50 model years – see below) so that the total inventory of Si is strictly conserved. A similar approach was adopted for DIC, Alk, DIN and $O_2$.

### 2.1.2  The sediment model MEDUSA2

MEDUSA is a time-dependent one-dimensional numerical model of coupled early diagenetic processes in sea-floor surface sediments. The original model version (MEDUSA v1) was described in Munhoven (2007). MEDUSA v1 has evolved to become MEDUSA2 which allows for a flexible chemical composition of the sediment, of the network of chemical transformations that describe the diagenetic processes (e.g. denitrification), and chemical equilibria to consider. It also offers a variety of Application Programming Interfaces (APIs) for coupling it to ocean models with different grid configurations and biogeochemical components (Munhoven, 2021). Here, we provide only a general description of the MEDUSA2 configuration used in this study; for details, including the exact equations and parameter values adopted, please refer to the technical report "MEDUSA Setup and Selected Configuration Options" in the Supplement.

In MEDUSA a sediment column is divided into three realms (the optional fourth one, a Diffusive Boundary Layer at the sediment-water-interface was not considered here). The topmost part from the sediment surface is called REACLAY and encompasses the reactive mixed-layer where solids sinking from the bottom layer of the ocean are collected. This is where chemical reactions take place, solids are transported by bioturbation and advection resulting from the continuous deposition of new material, and solutes by molecular diffusion. The second major realm is the located underneath, and is called CORELAY. It is made of a stack of sediment layers, typically $1\,\mathrm{cm}$ thick each. Here, no reactions or mixing take place: solids are buried and preserved in this realm which is building up a synthetic sediment core. REACLAY and CORELAY are connected by an thin transitional layer (TRANLAY) which acts as a short-term (numerical) storage buffer and which can also be seen as the topmost layer of CORELAY. In the MEDUSA2 configuration used here, sediment columns (one per seafloor grid element) resolve a $50\,\mathrm{cm}$ thick reactive surface sediment layer on a vertical grid with 71 points to take into account diagenetic processes acting at depths greater than $10\,\mathrm{cm}$, such as organic degradation by sulfate reduction. The grid point spacing is not regular but increases with depth in the sediment for a better representation of the strong subsurface solute concentration gradients. Only the uppermost $10\,\mathrm{cm}$ are mixed by bioturbation. There is no lateral exchange between sediment columns.

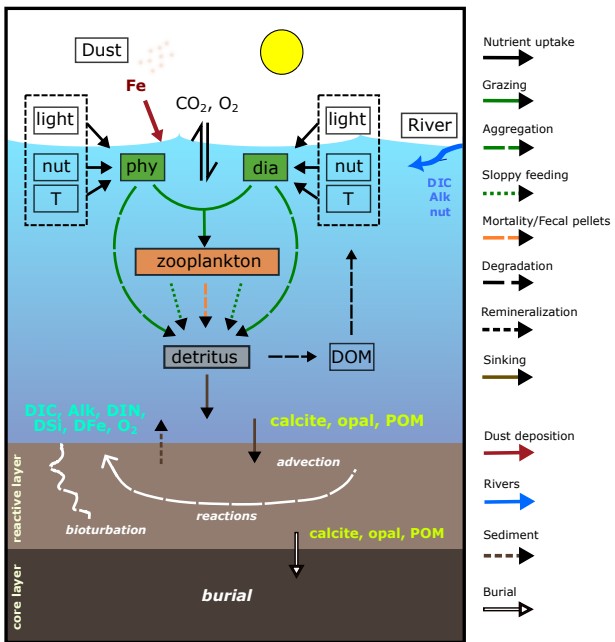

**Figure 1.** Schematic diagram of the components and interactions in REcoM3p coupled with the sediment model MEDUSA2 (modified and extended from Gürses et al. (2023, Fig. 2)). Small phytoplankton (**phy**) and diatoms (**dia**) take up inorganic nutrients (**nut**) and grow in dependence on **light** and temperature (**T**). One generic **zooplankton** consumes phytoplankton. Phytoplankton aggregation, zooplankton sloppy feeding, mortality and fecal pellets generate sinking **detritus**. Sinking detritus degrades to dissolved organic matter (**DOM**) which then remineralizes to dissolved inorganic carbon (DIC) and nitrogen (DIN). **Calcite**, **opal** and particulate organic matter (**POM**) reaching the seafloor enter the reactive layer of sediments, where accumulation, bioturbation, degradation and dissolution take place. Dissolved products of these processes (**DIC, Alk, DIN, DSi, DFe** and **$O_2$**) go back to the bottom water by diffusion. The solids accumulate and are buried further in the core layer. Sources of DIC, Alk and nutrients to the ocean include sediments and rivers and dust deposition is an additional source of iron.

MEDUSA has already been coupled to several ocean biogeochemistry and Earth System Models (Moreira Martinez et al., 2016; Kurahashi-Nakamura et al., 2020; Munhoven, 2021). Coupling to ocean models is done through so-called 'applica-
tions' in the MEDUSA code. We introduced a new application `medusa-fesom-recom` which controls 1) the reading of FESOM2.1-REcoM3p input and conversion into format and units that MEDUSA requires, 2) the selection of processes and global rate parameter values for tracing the evolution of the concentrations of solids and solutes considered, and 3) the writing of the resulting diffusive solute exchange with the ocean to a file for usage by FESOM2.1-REcoM3p and the obtained burial loss of solids into the sediment core. These burial losses can be used to monitor and/or regulate oceanic mass balances of carbon, alkalinity and the main nutrients (nitrogen and silicon).

Consistent with the input from FESOM2.1-REcoM3p, we chose a MEDUSA2 configuration with five solids (clay, calcite, opal and two types of organic matter) and nine solute components ($CO_2$, $HCO_3^-$, $CO_3^{2-}$, $O_2$, $NO_3^-$, $H_4SiO_4$, $NH_4^+$, $SO_4^{2-}$ and

HS$^-$). The two types of organic matter are needed to account for the variable stoichiometry in REcoM3p. Processes altering the content of solids and solutes in sediments include calcite dissolution, organic degradation by aerobic respiration, denitrification

and sulfate reduction, opal dissolution, and chemical equilibria of the carbonate system in the porewaters. Manganese and iron reduction are not considered for organic matter degradation since their contribution is negligible at the global scale (Thullner et al., 2009). REcoM3p only calculates formation and dissolution of calcite and does not represent aragonite. Correspondingly, only calcite dissolution in sediments is considered in the MEDUSA2 application `medusa-fesom-recom`.

As mentioned above, in the model setup organic matter can get degraded through aerobic respiration, nitrate and sulfate

reduction, i.e., organic matter is preserved and buried once it reaches a sediment depth which is devoid of $O_2$, $NO_3^-$ and $SO_4^{2-}$. Bottom water concentrations of $O_2$ and $NO_3^-$ are taken from FESOM2.1-REcoM3p output, while the $SO_4^{2-}$ concentration is derived from the bottom water salinity following Dickson et al. (2007) since FESOM2.1-REcoM3p does not explicitly represent sulfur. $NH_4^+$ and $HS^-$ concentrations at the sediment-water interface are set to 0 throughout the ocean (Thullner et al., 2009).

Biological components in REcoM3p have variable intracellular stoichiometry and thus the seafloor deposition fluxes of POC and PON (particulate organic nitrogen) have no fixed ratio. However, in MEDUSA2 degradation of particulate organic matters (POM) is calculated for POM classes with a fixed stoichiometry each. We therefore defined two end-member classes of POM in MEDUSA2 in which $Q = C : N$ is fixed with $Q_1 = 106 : 21$ and $Q_2 = 200 : 11$, respectively, representing the minimum and maximum C:N ratio simulated in the seafloor deposition flux in REcoM3p. The total outgoing fluxes of PON from REcoM3p

($F_N^o$) were then partitioned into two incoming contributions $F_N^{i1}$ and $F_N^{i2}$, according to

$$F_N^{i1} = \frac{Q_2 - Q}{Q_2 - Q_1} \cdot F_N^o \tag{1}$$

$$F_N^{i2} = \frac{Q - Q_1}{Q_2 - Q_1} \cdot F_N^o \tag{2}$$

where $Q = F_C^o / F_N^o$ is the ratio of the bulk POC flux ($F_C^o$) to the PON flux ($F_N^o$) that reaches the seafloor in REcoM3p. The carbon fluxes carried by the two POM classes are finally calculated by multiplying the nitrogen fluxes $F_N^{i1}$ and $F_N^{i2}$ with the

190 respective $C : N$ ratios:

$$F_C^{i1} = Q_1 \cdot F_N^{i1} \tag{3}$$

$$F_C^{i2} = Q_2 \cdot F_N^{i2} \tag{4}$$

The degradation time scale of organic matter depends on its elemental composition, i.e. the $C : N$ ratio (Amon and Benner, 1994; Martin et al., 1987). In the water column in FESOM2.1-REcoM3p, we considered a faster remineralization of nitrogen

compared to carbon with the ratio of 1.1:1 ($\rho_{DetN}$ and $\rho_{DetC}$ in Gürses et al. (2023), Tab. A8). The rate law expression chosen for the oxic degradation of organic matter in the sediment is more complex: it is linear in the concentration of organic matter in porewaters (with separate expressions for $[POM_1]$ and $[POM_2]$), and includes a Monod-type (hyperbolic) limitation with respect to the concentration of oxygen in the porewaters ($[O_2]$, supplementary material Sect. 3.2). Organic matter degradation through nitrate and sulfate reduction is described in a similar way but taking into account the inhibition by oxygen (supplemen-

tary material Sect. 3.3 and 3.4); organic matter oxidation by sulfate reduction is inhibited by oxygen and nitrate. We adopted

a 100-fold faster degradation rate for the low-C:N organic matter class ($k_{ox1}$ for $POM_1$) than for the high-C:N organic matter class ($k_{ox2}$ for $POM_2$) (Soetaert et al., 1996).

Besides organic matter, calcite and opal, the simulated sediment contains an inert component, which we refer to as 'clay' here for the sake of simplicity, and which is ultimately of continental origin. It stems from dust particles deposited over the sea surface and from terrestrial materials transported to the oceans by rivers. In our model setup, annual mean dust deposition from Albani et al. (2014) is considered as the oceanic clay input into sediments. This dust deposition distribution leads, however, to unrealistically low sedimentation (solid deposition) rates at seafloor depths shallower than 3000 m, typically by a factor of 20–50, but occasionally by more than 100, when compared to the empirical relationship of Middelburg et al. (1997). We therefore increased the deposition rate of lithogenic material ('clay') by $10^{-2.4-Z/1250} \times 0.1 \times 2650 \, \mathrm{kg \, m^{-2} \, yr^{-1}}$, where $Z$ is the depth below sea-level (in m), 0.1 is the volume fraction of solids close to the sediment surface (for a porosity of 0.9), and 2650 is the density of lithogenic material (in $\mathrm{kg \, m^{-3}}$). This way, the global distribution of seafloor sedimentation rates is in better agreement with the the empirical relationship of Middelburg et al. (1997). The resulting global distribution of clay input used in this study is shown in Fig. B1.

## 2.2 Coupling REcoM3p and MEDUSA2

FESOM2.1-REcoM3p and MEDUSA2 are sequentially coupled through file exchange. Sinking fluxes of POC, PON, opal ($SiO_2$) and calcite out of the bottom water boxes are saved as output files by FESOM2.1-REcoM3p and read as input files by MEDUSA2 (Fig. 1). Furthermore, MEDUSA2 requires information on temperature, salinity and concentrations of alkalinity (Alk), DIC, oxygen, and nutrients in the bottom-most ocean model box. Temperature and salinity enter thermodynamic calculations in the sediment model and the bottom water concentrations are used in the calculation of diffusive fluxes between sediment and water column.

FESOM2.1-REcoM3p reads diffusive fluxes of nutrients including dissolved inorganic nitrogen (DIN) and dissolved silicate (DSi or $H_4SiO_4$), DIC, Alk and oxygen from the MEDUSA2 output file (Fig. 1). DFe input from sediments is derived from the diffusive flux of DIN, using a fixed Fe:N ratio. Other quantities that are calculated by MEDUSA2 are the permanent burial of carbon, organic matter, opal and calcite in the sediment core. This output is used to monitor and compensate changes in the total mass balances of carbon and the other tracers in the ocean and reactive sediment. The burial loss is balanced by adding the same quantities as riverine input which is distributed over the surface ocean in the model by scaling it with the local river runoff from the forcing data.

## 2.3 Model setup

### 2.3.1 Model configuration

FESOM2.1 employs unstructured meshes with variable horizontal resolution. The default mesh of FESOM2.1-REcoM3 (COREII mesh) has about 127 000 surface nodes with a nominal average resolution of 1 degree and enhanced resolution in the equatorial belt and at high latitudes going up to 25 km (Gürses et al., 2023). For testing the coupling with MEDUSA2 a reduced

model resolution (PI mesh) is used here, containing 3140 surface nodes, corresponding to a median horizontal resolution of 260 km (Butzin et al., 2023). This configuration reduces computational costs and simplifies simulations over the time scale of thousands of years in order to approach deep ocean equilibrium and significant changes in marine sediments. MEDUSA2 is coupled to the bottom layers of the ocean model. Therefore the horizontal grid within MEDUSA2 is always the same as in the ocean model.

Vertically, the ocean is divided into 47 layers and the layer thickness ranges from 5 m in the surface to 250 m in the deep ocean. The full free-surface formulation (zstar) was used, allowing vertical movement of all layers, to ensure tracer conservation in FESOM (Scholz et al., 2019, 2022). In this study, the model was retuned for the coarser resolution by reducing the maximum thickness diffusivity of the Gent–McWilliams parameterisation from $3000 \, \mathrm{m^2 \, s^{-1}}$ (used in the default FESOM2.1) to $1000 \, \mathrm{m^2 \, s^{-1}}$.

### 2.3.2 Forcing and initial conditions

FESOM2.1 is initialised with January temperatures and salinities from the Polar Science Center Hydrographic Climatology (PHC3, updated from Steele et al. (2001)) and driven by annually repeated atmospheric fields using the Corrected Normal Year Forcing Version 2.0 (CORE-NYF.v2, Large and Yeager, 2009).

The initial value of $X_{\mathrm{CO_2}}$ is 284.3 ppm following Meinshausen et al. (2017). Alk and DIC are initialised from version 2 of the Global Ocean Data Analysis Project (GLODAPv2) data set (Lauvset et al., 2016), DIN and DSi from the Levitus World Ocean Atlas climatology of 2013 (Garcia et al., 2014) and oxygen from the Levitus World Ocean Atlas climatology of 2018 (Garcia et al., 2019). The initial DFe field is based upon output from the Pelagic Interaction Scheme for Carbon and Ecosystem Studies (PISCES) model (Aumont et al., 2015), as outlined in Gürses et al. (2023).

Dust input of iron at the sea surface is calculated based on monthly averages of dust deposition by Albani et al. (2014) with a weight percentage for iron of 3.5% and a solubility of 2%. The riverine DFe input is based upon de Baar and de Jong (2001), who estimate that the rivers transport $26 \, \mathrm{Gmol \, Fe}$ as DFe to the oceans each year. These authors assume that about 90% of this is lost by flocculation when river water gets mixed with seawater, which reduces the actual input to the ocean to $2.6 \, \mathrm{Gmol \, Fe \, yr^{-1}}$. However, depending on types of catchment areas and the concomitant input of organic material which may act as metal chelator, the river input of DFe can be significantly higher (Guieu et al., 1996; Krachler et al., 2005). We therefore tuned our model by varying the river input of DFe (assuming an upper limit of $26 \, \mathrm{Gmol \, Fe \, yr^{-1}}$) to get a reasonable distribution of DFe and of the simulated biological productivity, and finally adopted a total riverine DFe input of $5.2 \, \mathrm{Gmol \, Fe \, yr^{-1}}$. The river input of DFe is distributed at the sea surface by scaling with the river runoff, which is part of the CORE-NYF.v2 forcing.

### 2.4 Coupled simulations with FESOM2.1-REcoM3p-MEDUSA2

A coupled simulation starts with a spinup run with FESOM2.1-REcoM3p ($R_{\mathrm{spinup}}$), followed by the pre-charging of MEDUSA2 (Fig. 2). Subsequently, FESOM2.1-REcoM3p and MEDUSA2 are run alternately with a defined coupling frequency of 50 years ($R_{\mathrm{coupled}}$).

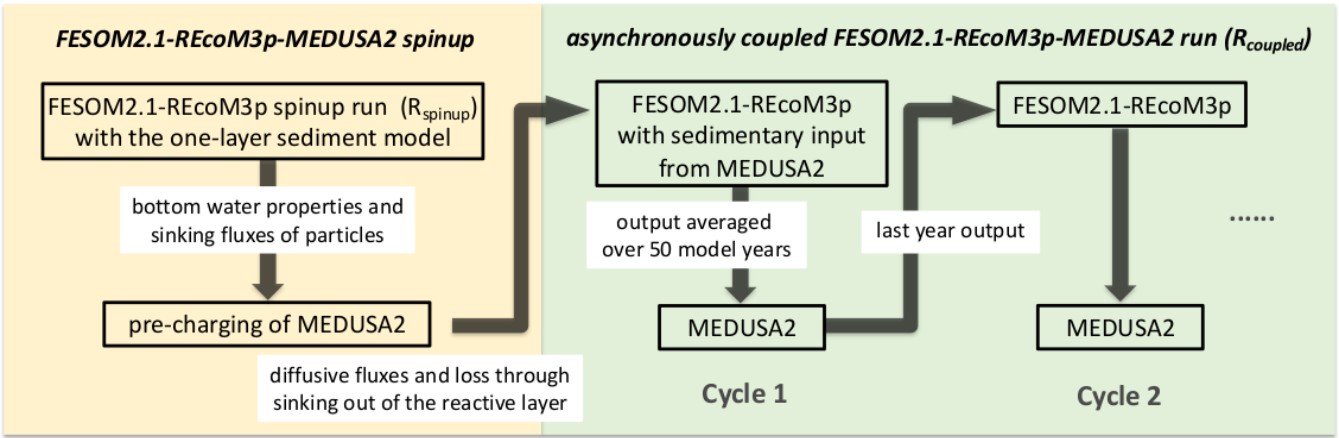

**Figure 2.** Workflow of a coupled FESOM2.1-REcoM3p-MEDUSA2 simulation.

### 2.4.1 FESOM2.1-REcoM3p spinup run ($R_{spinup}$)

FESOM2.1 (without biogeochemistry) was run for 1000 years as a spinup of the ocean circulation. After that, REcoM3p was switched on and run for another 1500 years to get a quasi-equilibrium of deep ocean concentrations. During these 1500 years, the exchange between ocean and sediment was calculated with the original one-layer sediment representation. Model output of the last 50 years was analysed as the initial conditions for $R_{coupled}$ in Section 3.1.

### 2.4.2 Pre-charging of MEDUSA2

Continuous exchange of material between ocean and sediments alters both ocean chemical boundary conditions and the content of the reactive sediment layer. The latter changes much more slowly due to low sedimentation rates. To reduce the computing time for getting significant changes in sediments, MEDUSA2 was first run for 100,000 years forced by the results from $R_{spinup}$ so that the sediment layers in MEDUSA are charged before an interactive coupled FESOM2.1-REcoM3p-MEDUSA2 simulation starts. This way, we may reach an initial seafloor sediment distribution that is as consistent as possible with the productivity pattern, the cycling in the water column and the boundary conditions prevailing at the seafloor (oxygenation, saturation state, etc).

### 2.4.3 Coupled simulation ($R_{coupled}$)

Two simulations were started from the state of year 1500 in $R_{spinup}$ and compared to demonstrate how carbon storage in sediments affects the marine carbon cycle and atmospheric $CO_2$ (Fig. 2): (1) $R_{sedbox}$ is the continuation of $R_{spinup}$ from year 1500 to year 3500; (2) a coupled simulation $R_{coupled}$ was conducted for 2000 model years starting with the precharged MEDUSA2 sediment layers (see previous section). A coupling frequency of 50 years was consistently applied between FESOM2.1-REcoM3p and MEDUSA2. For each coupling cycle, output of FESOM2.1-REcoM3p was averaged over 50 years before using it as input

in MEDUSA2. The sediment-to-ocean fluxes (input to FESOM2.1-REcoM3p) were updated every 50 years with results from the MEDUSA2 simulation. Within one coupling cycle the ocean-sediment exchange fluxes to be applied at each time step were kept constant. The outputs of $R_{sedbox}$ and $R_{coupled}$ were averaged over the last 50 years before comparison (Sect. 3.2).

### 2.4.4 Transient simulations with perturbations in atmospheric $CO_2$ ($R_{pert1k}$ and $R_{pert2k}$)

To demonstrate that the ocean-only setup of FESOM2.1-REcoM3p-MEDUSA2 can be used to study transient climate changes, two experiments were conducted starting from the final state of $R_{coupled}$, adding 1000 and 2000 PgC into the atmosphere, respectively. With those experiments, the interactions between the atmosphere, ocean and sediment under idealised ocean acidification scenarios were examined. Both coupled experiments ($R_{pert1k}$ and $R_{pert2k}$) were run for 2000 years and the temporal change in the atmospheric $CO_2$ concentration and calcite content in sediments was analysed.

### 2.4.5 Performance of the coupled model

FESOM coupled with REcoM spends about 80% of the total run time of a simulation on the tracer transport computations (Himstedt, 2023). An acceleration method was implemented for a parallel calculation of tracer advection and with two parallel tracer groups on 72 cores, a speedup by a factor of 1.8 was achieved for simulations with the reduced resolution using the PI mesh (Himstedt, 2023). In this study, each coupled FESOM2.1-REcoM3p-MEDUSA2 cycle (50 model years) is then completed within seven hours computation time on 72 cores, of which the MEDUSA2 related calculations require less than five minutes (i.e., of the order of 1% only).

## 3 Results and Discussion

### 3.1 FESOM2.1-REcoM3p spinup simulation with the one-layer sediment ($R_{spinup}$)

Generally, the global and basin-averaged profiles of DIC, Alk, DIN, DSi and $O_2$ in the FESOM2.1-REcoM3p spinup run ($R_{spinup}$) agree well with GLODAPv2 (Large and Yeager, 2009) and WOA data (Garcia et al., 2019) (Fig. A1), particularly in ocean basins covering large areas of the open ocean. The modelled $O_2$ concentration in the Arctic Ocean is clearly lower than observed. This will be discussed in Sect. 3.2.5 below.

The global net primary production (NPP) of 35 $PgC\,yr^{-1}$ is lower than the satellite-based estimates but comparable to other modelling studies (see Gürses et al., 2023, Table 3 and references therein), e.g. 24.5–57.3 $PgC\,yr^{-1}$ in CMIP6 (Séférian et al., 2020). The larger part of NPP comes from the small phytoplankton (23 $PgC\,yr^{-1}$); diatoms contribute the remaining 12 $PgC\,yr^{-1}$. Carbon export out of the upper 100 m into the deep ocean is 6.6 $PgC\,yr^{-1}$. The slightly higher productivity and export found here compared to an NPP of 32.5 $PgC\,yr^{-1}$ in the base version of FESOM2.1-REcoM3 (Gürses et al., 2023) can be explained by the differences between the model setups: 1) a much coarser spatial resolution of the PI mesh used here and a different forcing data set, which result in differences in resolved physical processes (e.g., circulation and mixing) and

**Table 1.** Seafloor deposition and burial fluxes of POC ($PgC\,kyr^{-1}$), calcite ($PgC\,kyr^{-1}$) and opal ($Pmol\,Si\,kyr^{-1}$) in simulations and observation-based estimates, reported for the global ocean and ocean regions deeper than 1 km. $R_{high}$ is shown here to demonstrate the impact of model resolution on the simulated seafloor deposition and this study rather focuses on the low-resolution simulations.

| | Seafloor deposition | | | | | |
| --- | --- | --- | --- | --- | --- | --- |
| | POC | | calcite | | opal | |
| | global | > 1 km | global | > 1 km | global | > 1 km |
| $R_{sedbox}$ | 650 | 410 | 380 | 370 | 70 | 65 |
| $R_{coupled}$ | 420 | 270 | 370 | 360 | 80 | 70 |
| $R_{high}$ | 1380 | 415 | 300 | 260 | 90 | 70 |
| Observed | 930–5739[a] | 310–1029[b] | | | 22–40[c] | 79–84[d] |

| | Burial | | | | | |
| --- | --- | --- | --- | --- | --- | --- |
| | POC | | calcite | | opal | |
| | global | > 1 km | global | > 1 km | global | > 1 km |
| $R_{coupled}$ | 86 | 28 | 100 | 90 | 18 | 13 |
| Observed | 160–2600[e] | 2–300[f] | 280[g] | 100–150[h] | 7.1[i] | 5.9–9.2[j] |

Reference keys: B: Burdige (2007), C16: Cartapanis et al. (2016), C18: Cartapanis et al. (2018), D07: Dunne et al. (2007), D12: Dunne et al. (2012), Ha: Hayes et al. (2021), Hi: Hilton and West (2020), J: Jahnke (1996), M: Muller-Karger et al. (2005), N: Nelson et al. (1995), Sa: Sarmiento et al. (2002), Se: Seiter et al. (2005), T95: Tréguer et al. (1995), T13: Tréguer and De La Rocha (2013), T21: Tréguer et al. (2021).

[a] B, D07, M, Sa; [b] B, J, M, Sa, Se; [c] D07, N, T95; [d] J, T21; [e] B, C18, D07, M; [f] B, C16, C18, D07, Hi, Ha, J, M, Se; [g] C18; [h] C18, D12, Ha, Sa; [i] T95; [j] Ha,T13,T95,T21

thus in the environmental conditions for phytoplankton growth (e.g., light, temperature and nutrient supply); 2) REcoM3p uses a configuration with a single zooplankton class whereas the simulations in Gürses et al. (2023) contained two zooplankton classes; 3) additional iron input from rivers relieves iron limitation of phytoplankton growth in some regions.

Deposition fluxes from the ocean bottom layer onto the top of the sediments from different simulations and burial fluxes of POC, calcite and opal from the coupled simulation are summarized in Table 1 along with observation-based estimates. The simulated global deposition rate of POC ($650\,PgC\,kyr^{-1}$) in $R_{spinup}$ is lower than the range of observation-based estimates ($930–5739\,PgC\,kyr^{-1}$) reported by Burdige (2007). This is not surprising since the global primary and export production in our model are both lower than observations. The simulated POC deposition rates (Fig. 3a) are in the same order of magnitude as Dunne et al. (2007) but mainly occur on top of deep-sea sediments (deeper than 1 km). The contribution of the deposition rates in shallower waters (37%) underestimates the relative share of 67–82% obtained by others (Muller-Karger et al., 2005; Burdige, 2007). Models with a coarse resolution do not resolve physical processes and thus the biological recycling of carbon in shelf regions well, likely leading to an unrealistic estimation of POC sinking into and accumulation in sediments. To quantify

the effect of model resolution, we did a simulation with exactly the same model code and setup but at a higher resolution with 126,858 surface nodes ($R_{high}$), i.e., as in Gürses et al. (2023). The total POC deposition rate of 1380 $PgC\,kyr^{-1}$ in $R_{high}$ fits in the estimated range, and the flux in shallower waters represents a larger fraction (70%) of the global flux which falls within the range of estimates (67–82%). Here, we still consider model results with a coarse resolution which is commonly used for

technical tests, allowing us to run a reasonable number of tuning experiments and coupled simulations over several thousands of years within a realistic time frame.

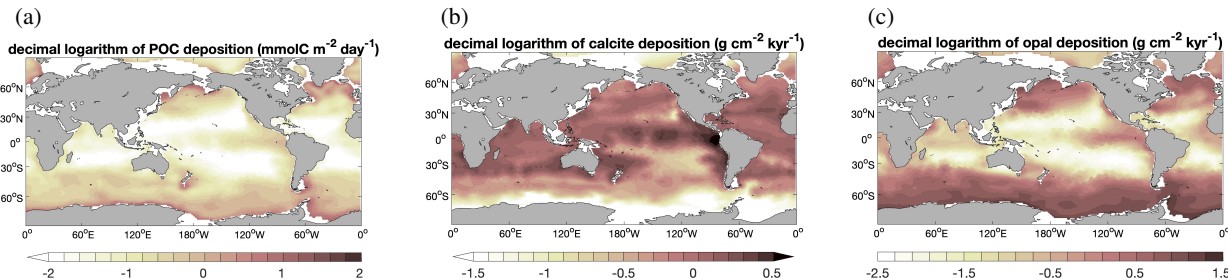

**Figure 3.** Decimal logarithm of seafloor deposition rates of (a) POC ($mmol\,C\,m^{-2}\,day^{-1}$, i.e., the same units as in Dunne et al. (2007)), (b) calcite ($g\,cm^{-2}\,kyr^{-1}$) and (c) opal ($g\,cm^{-2}\,kyr^{-1}$) in $R_{spinup}$.

A total calcite deposition rate of 380 $PgC\,kyr^{-1}$ is found to reach the ocean-sediment interface in $R_{spinup}$, from which 370 $PgC\,kyr^{-1}$ happened in the deep ocean below 1 km water depth. Be aware that the omissions of aragonite and of the benthic production of $CaCO_3$ (e.g., by coral reefs) are important shortcomings of our approach. Buitenhuis et al. (2019)

simulated three pelagic calcifiers and estimated a contribution of aragonite producers to shallow water export of $CaCO_3$ at 100 m of at least 33%. Furthermore, coccolithophore and calcifying zooplankton together are reported to contribute to the global carbonate fluxes by 40–60% and the rest of the fluxes remains unexplained (Knecht et al., 2023), which also results in high uncertainty in the simulating calcifying organisms and $CaCO_3$ fluxes. Our model roughly reproduces the spatial pattern of the Th-normalized deposition fluxes (Hayes et al., 2021) with high fluxes in the North Atlantic and Arabian Sea (up to

2 $g\,cm^{-2}\,kyr^{-1}$), lower fluxes in large areas in the Pacific and Southern Ocean ($< 1\,g\,cm^{-2}\,kyr^{-1}$, Fig. 3b). Only some parts of the eastern equatorial Pacific region, the modelled calcite fluxes are about two times higher than in Hayes et al. (2021) which might be caused by the too high calcite production in this region in the model.

The seafloor deposition rate of opal in $R_{spinup}$ is 70 $Pmol\,Si\,kyr^{-1}$, of which 65 $Pmol\,Si\,kyr^{-1}$ take place at seafloor depths greater than 1 km. Observation-based estimates available in the literature unfortunately provide a conflicting picture, with

22–40 $Pmol\,Si\,kyr^{-1}$ for the total flux (Dunne et al., 2007; Nelson et al., 1995; Tréguer et al., 1995), while in the ocean that is deeper than 1 km $84 \pm 17\,Pmol\,Si\,kyr^{-1}$ (Tréguer and De La Rocha, 2013; Tréguer et al., 2021) should settle. Assuming that the more recent data are of better quality, we conclude that the simulated opal deposition rates in the deep ocean agree well with reconstructions while for the total rates a revised data set seems necessary. The spatial distribution of opal fluxes agree qualitatively well with Hayes et al. (2021) with high fluxes at high latitudes in both hemispheres and moderate ones in

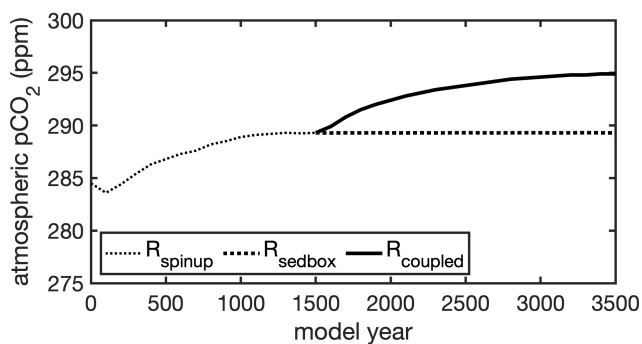

**Figure 4.** Simulated atmospheric $CO_2$ during 3500 model years: a FESOM2-REcoM3p spinup simulation with an integrated one-layer sediment ($R_{spinup}$) was run for 1500 years; after 1500 years the model with an integrated one-layer sediment was run for further 2000 years ($R_{sedbox}$) and a simulation coupled with MEDUSA2 ($R_{coupled}$) was branched off and run for 2000 years.

the eastern equatorial Pacific (Fig. 3c). However, the model shows much higher values in the Southern Ocean, indicating that the Fe limitation of diatom growth in the Southern ocean is too weak in the model.

During the total 1500 simulated years the atmospheric $CO_2$ concentration first rises with time and reaches 289 ppm at the end of $R_{spinup}$ (Fig. 4).

## 3.2   The coupled simulation with FESOM2.1-REcoM3p-MEDUSA2 ($R_{coupled}$)

### 3.2.1   Sediment content

The weight percentage of sediment composition (Fig. 5) is compared in the following with the data compilation of the surface sediment composition of Hayes et al. (2021). It should be noted that this latter broadly agrees with the alternative and older compilation of Seiter et al. (2004).

Simulated calcite content in $R_{coupled}$ (Fig. 5, top row) exhibits high values (up to >80%) in the Atlantic, tropical and sub-
tropical South Pacific as well as the Indian Ocean, and lower values (near zero) in the North Pacific and the Southern Ocean. Also, the calcite-rich sediments along the Atlantic mid-ocean ridge are reproduced to some extent in the model. This simulated pattern generally agrees well with Hayes et al. (2021).

Opal content (Fig. 5, middle row) is elevated at high latitudes in the North Pacific and North Atlantic Ocean, as well as in the Southern Ocean at the Antarctic Polar Front. This is also seen in the data compilation. The opal distribution mainly reflects the
diatom productivity (Fig. 6a) and opal deposition rates (Fig. 3). The latter has a similar pattern as [230]Th-normalized estimates by Hayes et al. (2021), whereas much higher fluxes are found in the model over large areas in the Southern Ocean. This could lead to a likely overestimation of opal content in sediments, although not many observations are available for these areas. The opal belt in the equatorial eastern Pacific is smaller and less pronounced in the model than observed. This is related to the somewhat too strong iron limitation of diatoms in this region in our model.

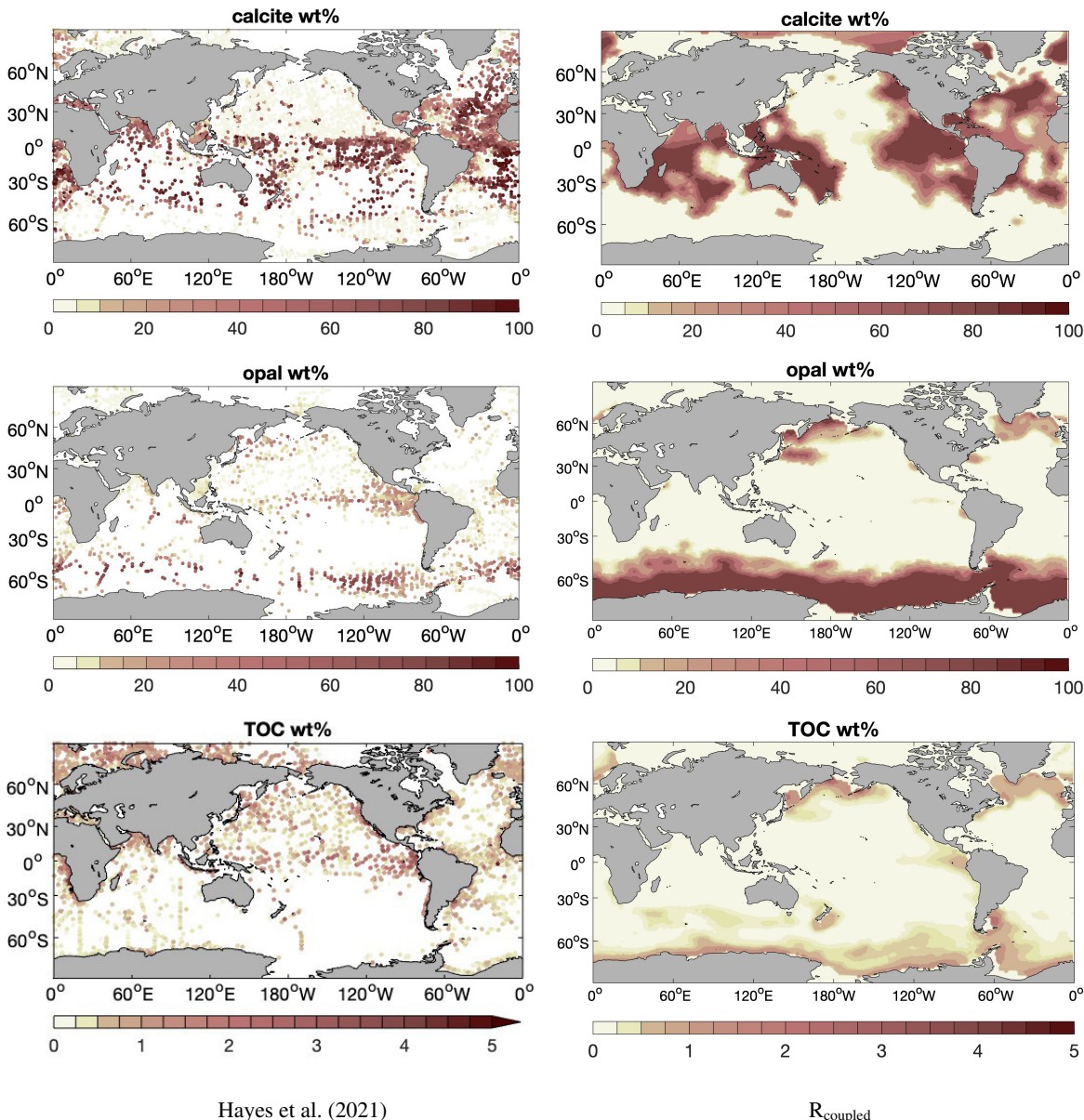

**Figure 5.** Distribution (weight %) of (top row) calcite, (middle row) opal and (bottom row) total particulate organic carbon (TOC) in the sediment, averaged over the upper 10 cm of sediments. Left: data compilation of averages over the Holocene age and measurements reported for the surface sediment by Hayes et al. (2021); Right: results from simulation $R_{coupled}$.

Simulated sediment TOC (Fig. 5, bottom row) is elevated at high latitudes in the Atlantic Ocean, North Pacific and Southern Ocean, similar to opal. Also significantly higher TOC preservation is found in the eastern equatorial Pacific. Beside the contribution by diatoms, small phytoplankton in the model also has a high productivity in this region (Fig. 6b). Only a small

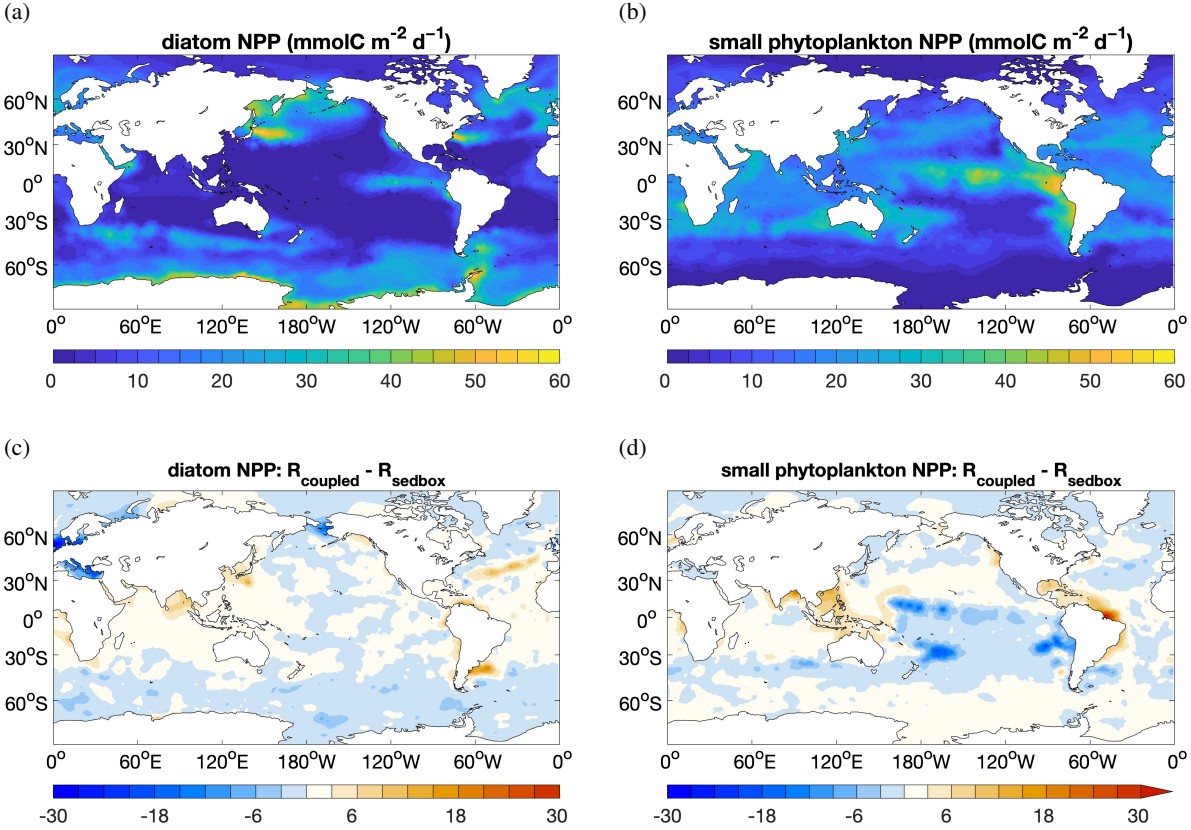

**Figure 6.** NPP $(\mathrm{mmolC\,m^{-2}\,day^{-1}})$ of (a) diatoms, and (b) small phytoplankton in $R_{sedbox}$ and the difference in NPP (c, d) between the two simulations ($R_{coupled}$-minus-$R_{sedbox}$).

amount of TOC is present in sediments in large areas of the open ocean. The global pattern of sediment TOC content roughly agrees with data compilation, although a detailed comparison of POC content in the southern hemisphere, particularly in the
South Pacific Ocean, is not possible due to lack of data. The magnitude of TOC preservation in shallow waters and upwelling regions is somewhat lower compared to data compilation. This may in part be explained by the fact that the modelled biological production and thus the deposition flux to sediments are both lower than observation-based estimates (Sect. 3.1).

### 3.2.2 Degradation of organic matter in sediments

Three different pathways of degradation of organic matter in sediments are considered here: aerobic respiration, nitrate reduc-
tion and sulfate reduction. This setup offers the possibility to have a closer look at their roles in different ocean regions. Figure 7 shows the logarithm of organic carbon degradation rate $(\mathrm{\mu molC\,cm^{-2}\,yr^{-1}})$ by aerobic respiration (a), nitrate reduction (b) and sulfate reduction (c), integrated over the upper sediment layers of 10 cm. Aerobic respiration and denitrification roughly follow the pattern of POC deposition flux (Fig. 3a), while sulfate reduction mainly concentrates in much smaller areas at

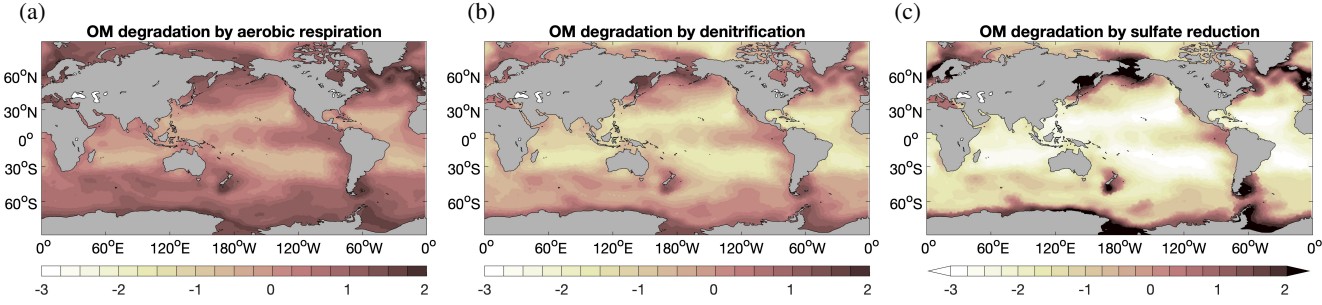

**Figure 7.** Decimal logarithm of organic carbon degradation rates ($\mu molC\,cm^{-2}\,yr^{-1}$) by aerobic respiration (a), denitrification (b) and sulfate reduction (c). The rates are vertically integrated over the top 10 cm of the modelled reactive sediment layer, i.e., the mixed layer.

high-latitudes and some upwelling regions with high biological productivity. In large areas of the deep-sea sediments, aerobic
385 respiration is the dominant degradation process. Nitrate reduction has lower rates than aerobic respiration in most regions of
the world ocean, except for the high-latitude North Pacific where porewater oxygen is fully consumed through organic matter
degradation. In high-latitude regions, high rates of sulfate reduction reach up to two orders of magnitude of those of the other
two processes.

 Globally in the modelled $50\,cm$ reactive sediment layer, about $33\,TmolC\,yr^{-1}$ is remineralized, where 45% is contributed by
390 aerobic respiration, 9% by nitrate and 46% by sulfate reduction. Our total carbon remineralisation is comparable to Sarmiento
and Gruber (2006) ($\sim 27\,TmolC\,yr^{-1}$), while previously reported data-based estimates and model results cover a large range
from 19 to $260\,TmolC\,yr^{-1}$ (Thullner et al., 2009; Burdige, 2007; Smith and Hollibaugh, 1993; Jørgensen, 1983).

 Aerobic respiration of $15\,TmolC\,yr^{-1}$ falls below the range of estimates based on oxygen consumption ($33$–$97\,TmolC\,yr^{-1}$)
for deep-sea sediments ($> 1000\,m$)(Jahnke, 1996; Christensen, 2000; Andersson et al., 2004; Seiter et al., 2005; Glud, 2008;
395 Jørgensen et al., 2022), and is much lower than estimates for the global sediments ($99$–$212\,TmolC\,yr^{-1}$, see Snelgrove et al.,
2018; Stratmann et al., 2019; Jørgensen et al., 2022) but substantially higher than some model results (e.g. $3.1\,TmolC\,yr^{-1}$ by
Thullner et al. (2009)).

 Denitrification removes about $2.9\,TmolN\,yr^{-1}$ is within the range of previous estimates of $1$–$12\,TmolN\,yr^{-1}$ (DeVries
et al., 2013; Thullner et al., 2009; Liu and Kaplan, 1984; Hattori, 1983; Jørgensen, 1983; Codispoti and Christensen, 1985;
400 Christensen, 1994; Christensen et al., 1987) but lower than the range of $16$–$20\,TmolN\,yr^{-1}$ of Middelburg et al. (1996).

 Sulfate reduction accounts for 46% of the global carbon mineralisation rate in our model, within the range between 30–76%
reported in previous studies (Canfield et al., 2005; Jørgensen and Kasten, 2006; Thullner et al., 2009). The highest values of
sulfate reduction are around $400\,\mu molC\,cm^{-2}\,yr^{-1}$, in line with the data compilation by Middelburg et al. (1997) for sediments
in shallower waters.

### 3.2.3 Solute exchange across the sediment-water interface

The diffusive flux of DIC from the sediment to the ocean shows a similar pattern to DIN, with high fluxes in regions with high input of organic matter into sediments (Fig. 8a and c). One exception for DIN is the net flux of DIN from the ocean into the sediment in regions along the Pacific coasts. In Fig. 7 these regions are characterised by high rates of denitrification, which results in a substantial reduction of DIN in the porewater and thus a net diffusion of DIN from the ocean bottom water to the sediment.

Diffusive fluxes of $O_2$ show more or less the opposite pattern to DIC. In regions where the seafloor deposition rate of organic matter is high (Fig. 8d), e.g., in the Northern Hemisphere around $60°$ or in the Southern Ocean, degradation of organic matter leads to a high $O_2$ flux from the ocean to the sediments as well as high DIC flux from the sediment to the ocean.

The Alk flux distribution (Fig. 8b) looks more complex and is the result of two processes that have opposite effects: degradation of organic matter decreases the alkalinity in porewater, while calcite dissolution increases it. Therefore, in those regions where the organic matter degradation rate in the surface sediment is high (i.e., where $O_2$ uptake is high – Fig. 8d) alkalinity in porewaters may get lowered to the extent that there is a net influx of alkalinity from the ocean bottom water to the sediment. In the Atlantic, the Indian Ocean and parts of the Pacific Ocean where calcite inputs to the sediment are high, alkalinity in porewater is clearly increased and there is a net diffusive flux of alkalinity out of the sediment, into the ocean bottom water.

### 3.2.4 Burial fluxes out of the reactive layer

The simulated POC burial flux in the global sediment ($86\,\mathrm{PgC\,kyr^{-1}}$) is lower than the observed range ($160\text{–}2600\,\mathrm{PgC\,kyr^{-1}}$, Tab. 1), consistent with the comparison for the productivity and sinking fluxes. In the deep-sea sediments the simulated flux ($28\,\mathrm{PgC\,kyr^{-1}}$) is within but close to the lower end of the observed range ($2\text{–}300\,\mathrm{PgC\,kyr^{-1}}$), reflecting again the inability of our model to represent shallow-water processes with the current resolution.

Similarly, the simulated global burial flux of $CaCO_3$ ($100\,\mathrm{PgC\,kyr^{-1}}$) is much lower than the observation-based estimate ($280\,\mathrm{PgC\,kyr^{-1}}$), while the deep-sea burial of $95\,\mathrm{PgC\,kyr^{-1}}$ is close to the lower end of the observed range of $100\text{–}150\,\mathrm{PgC\,kyr^{-1}}$. The observation-based estimates suggest a roughly equal distribution between shallow and deep-sea environments, while the model simulates only about 5% of the global calcite burial in sediments at depths shallower than $1\,\mathrm{km}$. The possible causes is the omission of some $CaCO_3$ producers in REcoM3p which has been already discussed in Sect. 3.1.

The simulated opal burial in deep-sea sediments ($13\,\mathrm{Pmol\,Si\,kyr^{-1}}$) exceeds the observed range ($5.9\text{–}9.2\,\mathrm{Pmol\,Si\,kyr^{-1}}$), reflecting an overestimation of opal deposition in large areas in the Southern Ocean. It is difficult to compare the modelled global burial of $18\,\mathrm{Pmol\,Si\,kyr^{-1}}$ with the only available but relatively old estimate of $7.1\,\mathrm{Pmol\,Si\,kyr^{-1}}$ by Tréguer et al. (1995) which is even lower than other estimates for the deep-sea burial. This issue has been already mentioned by discussing opal deposition flux in Sect. 3.1.

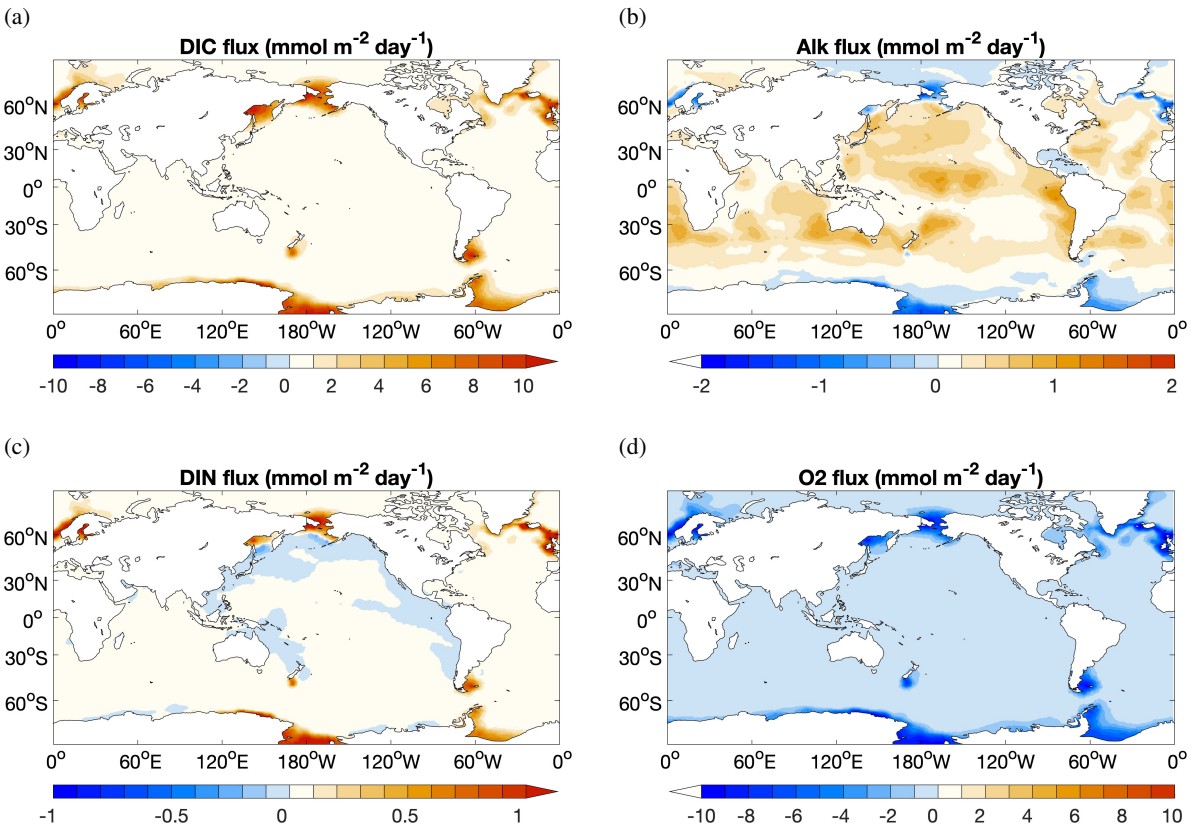

**Figure 8.** Diffusive flux of DIC, Alk, DIN and $O_2$ from the sediment to the ocean ($\mathrm{mmol\,m^{-2}\,day^{-1}}$). Sources for the ocean are shown as positive values.

### 3.2.5 Impact of the complex sediment on productivity and nutrient supply

The globally averaged vertical distributions of DIC, Alk, $O_2$ and nutrients do not differ much between $R_{sedbox}$ (Fig. A1) and $R_{coupled}$ (Fig. 9). The $O_2$ distribution in the Arctic Ocean is considerably improved in $R_{coupled}$. In MEDUSA2, the oxic degradation rate of organic matter depends on the $O_2$ concentration in the sediment, whereas in the one-layer sediment model, $O_2$ consumption in sediments is calculated with a fixed $O_2$:C ratio and subtracted from the bottom water $O_2$ concentration, likely leading to an overestimation of degradation of organic matter and the lowering of $O_2$ concentrations in the bottom water. This also applies to other regions with high deposition of organic matter such as the Southern Ocean and parts of the Pacific Ocean which also show small improvements in the $O_2$ profiles.

The marine NPP in the coupled simulation $R_{coupled}$ is nearly the same as in $R_{sedbox}$. The spatial distribution of NPP differences between the two simulations (Fig. 6) reveals higher productivity by both diatoms and small phytoplankton in coastal regions with large riverine nutrient inputs (DIN and DSi, Fig. 10c and d), which were not considered for $R_{sedbox}$.

Nutrient supply in the simulations using MEDUSA2 or the one-layer sediment differs in two ways. First, the total diagenetic flux of nutrients from the sediment to the ocean is lower when using MEDUSA2 (Fig. 10a and b; Table 2), since particles sinking into sediment can be stored there: a part is degraded or dissolved in the reactive layer and the remineralization products released to the porewaters from where they may diffuse back to the overlying ocean bottom waters, while the rest is buried in the deeper core layers (Munhoven, 2021). This storage and burial delay nutrient recycling and reduce the sedimentary nutrient source when compared to the full degradation and dissolution which takes places in the single-layer sediment. Second, the current riverine source of nutrients considered in the coupled simulation is estimated from the solid burial flux that leaves the reactive sediment layer to be transferred to the core layer in MEDUSA2. This additional source brings nutrients directly into surface waters near river mouths (Fig. 10c and d). As a result, diatom productivity shows a clear decrease in the North Sea and the Bering Sea (Fig. 6c) where DIN, DSi and DFe from sediments are all significantly reduced (Fig. 10a and b) and no riverine input can cover the loss (Fig. 10c and d).

In $R_{coupled}$, $16.8\,\mathrm{Tmol\,Si\,yr^{-1}}$ are delivered by rivers, while the sedimentary source only decreases from $73.2\,\mathrm{Tmol\,Si\,yr^{-1}}$ in $R_{sedbox}$ to $62.9\,\mathrm{Tmol\,Si\,yr^{-1}}$ (Table 2). On the other hand, the riverine input of $0.1\,\mathrm{Tmol\,N\,yr^{-1}}$ cannot compensate the decline in the sediment input from $8.1$ to $5.0\,\mathrm{Tmol\,N\,yr^{-1}}$. The nutrients supplied by rivers are, however, directly available for phytoplankton living in surface waters and can still induce phytoplankton growth in areas adjacent to river mouths (Fig. 6c and d), particularly in regions where sedimentary input does not change much (e.g. tropical and subtropical regions). The sedimentary source of iron strongly decreases as well (Table 2), however, the intensity of iron limitation for phytoplankton does not change significantly, since the riverine source is much higher and covers most of the regions where sedimentary input becomes smaller in $R_{coupled}$.

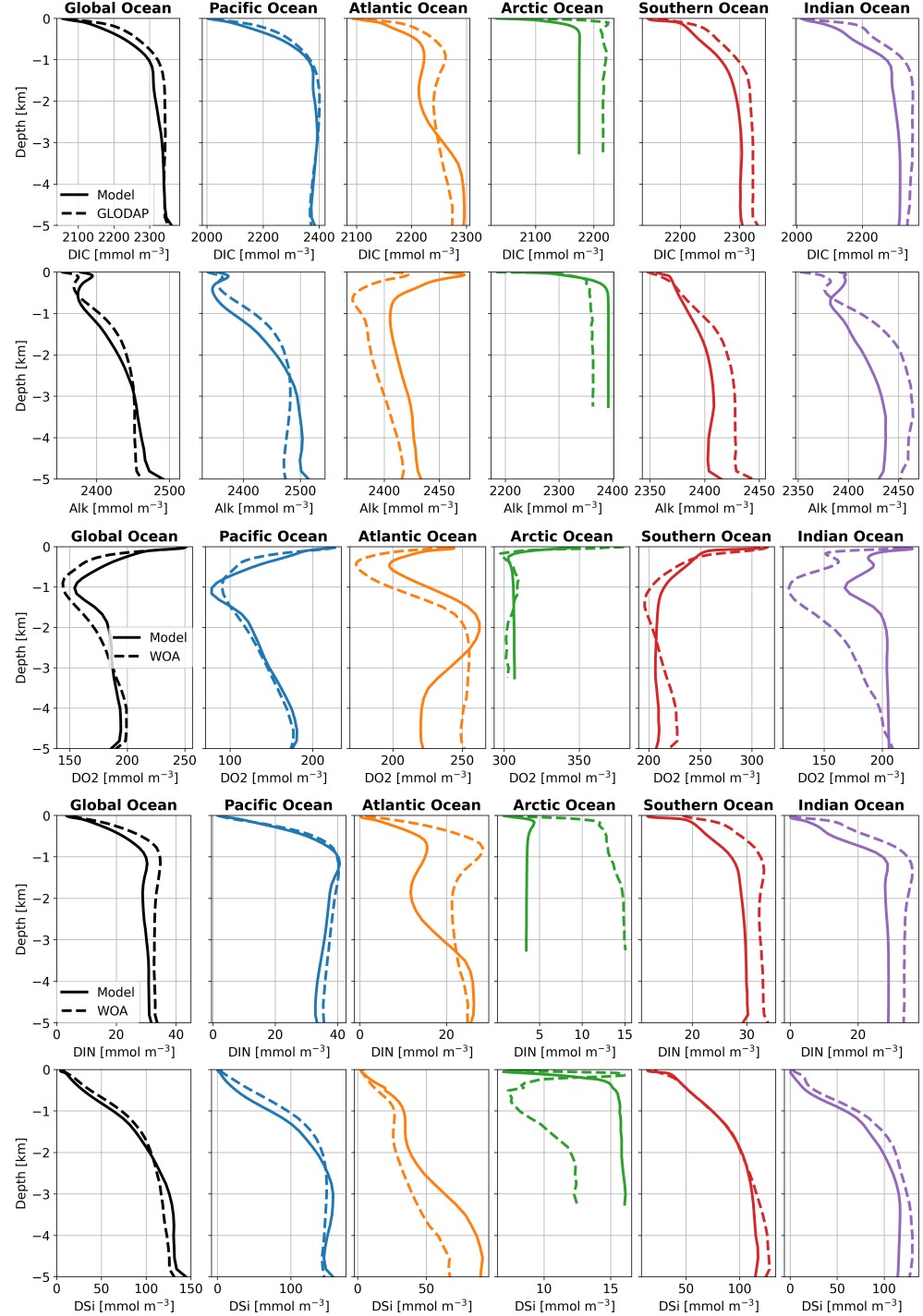

**Figure 9.** Averaged vertical profiles of DIC, Alk, $O_2$, DIN and DSi in ocean basins $(\mathrm{mmol\,m^{-3}})$ in $R_{coupled}$, compared with GLODAP and WOA data which were used as initial conditions in simulations in this study.

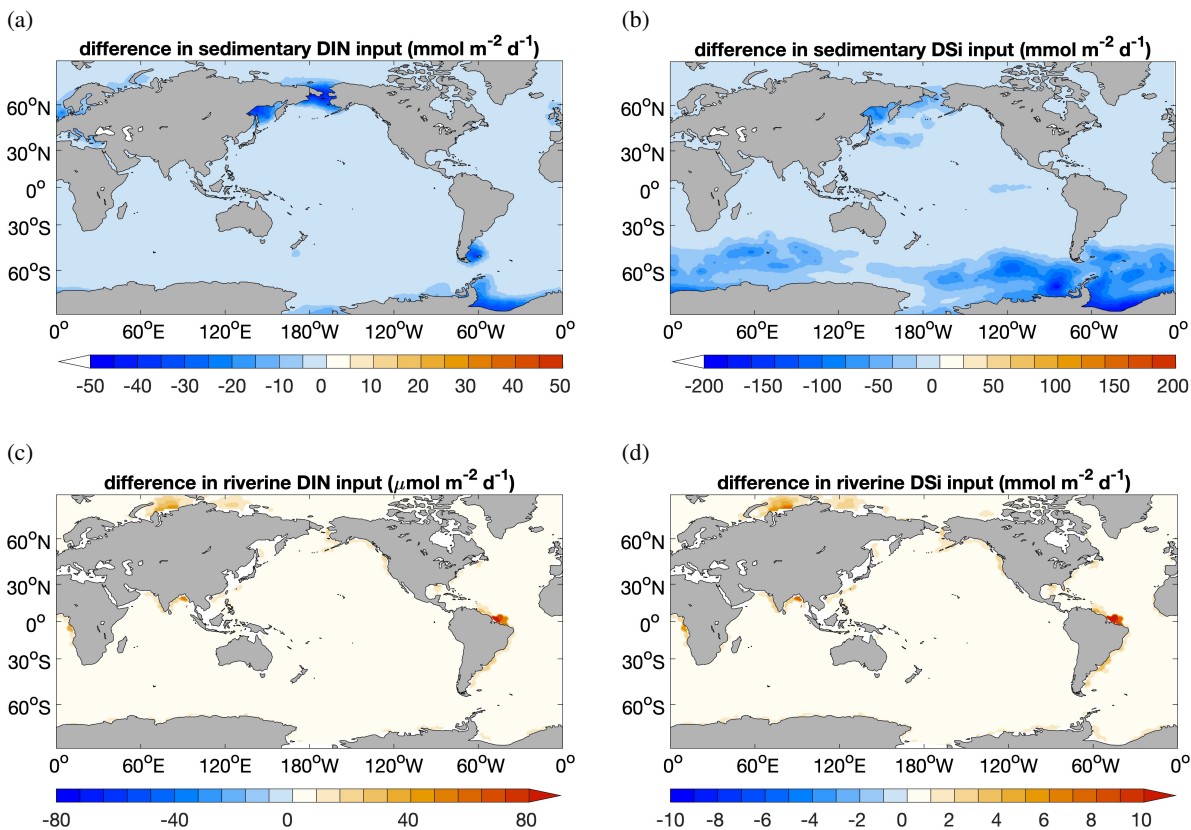

**Figure 10.** Decrease of sedimentary input of (a) DIN ($\text{mmol m}^{-2}\,\text{d}^{-1}$) and (b) DSi ($\text{mmol m}^{-2}\,\text{d}^{-1}$) in $R_{coupled}$ compared to $R_{sedbox}$; additional riverine input of (c) DIN ($\mu\text{mol m}^{-2}\,\text{d}^{-1}$) and (d) DSi ($\text{mmol m}^{-2}\,\text{d}^{-1}$) in coupled simulations $R_{coupled}$ ($\text{mmol m}^{-2}\,\text{d}^{-1}$). Change in sedimentary input of DFe has an identical spatial pattern as DIN since the iron source is calculated based on DIN source with a constant Fe:N ratio.

**Table 2.** Fluxes averaged over the last 50 years of the simulations. Diffusive flux of iron from sediments to the ocean is derived from the diffusive flux of DIN, using a fixed Fe:N ratio. Burial flux is calculated for the bottom of the reactive sediment layer at 50 cm. Positive fluxes are into the ocean or into sediments. Continued on next page. Note that the units here are $\mathrm{Tmol\,yr^{-1}}$, not $\mathrm{Pg\,yr^{-1}}$.

| | | $\mathrm{R_{sedbox}}$ | $\mathrm{R_{coupled}}$ |
|---|---|---|---|
| | Ocean balance | | |
| | riverine input | 0 | +9.0 |
| | diffusive flux out of sediment | +85.8 | +52.9 |
| | seafloor deposition (POC) | −54.2 | −35.5 |
| C | seafloor deposition (Calc) | −31.5 | −30.9 |
| | air-sea gas exchange | −0.2 | +0.3 |
| (Tmol year$^{-1}$) | Sediment balance | | |
| | seafloor deposition (POC) | +54.2 | +35.5 |
| | seafloor deposition (Calc) | +31.5 | +30.9 |
| | diffusive flux out of sediment | −85.8 | −52.9 |
| | burial (POC) | 0 | −1.9 |
| | burial (Calc) | 0 | −7.1 |
| | Ocean balance | | |
| | riverine input | 0 | +14.0 |
| | diffusive flux out of sediment | +47.9 | +41.8 |
| Alk | seafloor deposition (PON) | +9.0 | +5.6 |
| | seafloor deposition (Calc) | −62.9 | −61.8 |
| (Tmol year$^{-1}$) | Sediment balance | | |
| | seafloor deposition (PON) | −9.0 | −5.6 |
| | seafloor deposition (Calc) | +62.9 | +61.8 |
| | diffusive flux out of sediment | −47.9 | −41.8 |
| | burial (POM) | 0 | +0.1 |
| | burial (Calc) | 0 | −14.1 |

|  | Ocean balance | $R_{sedbox}$ | $R_{coupled}$ |
|---|---|---|---|
|  | riverine input | 0 | +0.1 |
|  | diffusive $NO_3$ flux out of sediment | +8.1 | +5.0 |
| N | seafloor deposition | −8.0 | −5.2 |
|  | Sediment balance |  |  |
| (Tmol year$^{-1}$) | seafloor deposition | +8.0 | +5.2 |
|  | diffusive $NO_3$ flux out of sediment | −8.1 | −5.0 |
|  | burial (PON) | 0 | −0.1 |
|  | Ocean balance |  |  |
|  | riverine input | 0 | +16.8 |
| Si | diffusive flux out of sediment | +73.2 | +62.9 |
|  | seafloor deposition | −72.4 | −80.4 |
| (Tmol year$^{-1}$) | Sediment balance |  |  |
|  | seafloor deposition | +72.4 | +80.4 |
|  | diffusive flux out of sediment | −73.2 | −62.9 |
|  | burial (opal) | 0 | −16.8 |
| Fe | dust | +5.8 | +5.8 |
| (Gmol year$^{-1}$) | rivers | +5.2 | +5.2 |
|  | diffusive flux out of sediment | +1.2 | +0.3 |

### 3.2.6 Impact of the complex sediment representation on atmospheric $CO_2$ and carbon storage

The oceanic carbon pools evolved towards equilibrium concentrations during $R_{coupled}$ by adjusting the gas exchange and the fluxes between ocean and sediment. The atmospheric $CO_2$ in $R_{coupled}$ increased to 295 ppm after 2000 years which is higher than the pre-industrial value of 284.3 ppm used to initialize the model, but consistent with the climate state determined by the CORE-NYF.v2 forcing. The air-sea gas exchange is not completely balanced at the end of the run with a net positive $CO_2$ flux from the atmosphere to the ocean of $0.3\,\mathrm{TmolC\,yr}^{-1}$ (Table 2), indicating that the atmosphere–ocean–sediment system has not yet reached its equilibrium. This can be seen in the temporal development of $CO_2$ (Fig. 4) and change in fluxes into and out of the sediment over time (Fig. C1).

We quantified the size of the carbon storage in the reactive sediment layer at the end of $R_{coupled}$, being aware of that the system is still in a transient state. Compared to $R_{sedbox}$, the ocean contains about 150 Pg less DIC. About 1390 PgC is accumulated in the sediment surface layer in $R_{coupled}$, mainly as calcite but with a 10% contribution from POC (Table 3). Emerson and Hedges (1988) estimated a POC storage of 150 PgC in the mixed layer of sediments and Parameswaran et al. (2024) recently reported that their modelled upper 10 cm of oceanic sediments harbors approximately 171 Pg TOC, while Archer (1996) reported that 800 PgC is stored as calcium carbonate within the 10 cm thick bioturbated layer. Our simulated carbon storage in the surface sediment (130 PgC as POC and 1060 PgC as calcite) is comparable with these observation- and model-based estimates. In $R_{sedbox}$, POC and calcite are almost completely degraded and dissolved in the single-layer sediment and thus the reservoir sizes of carbon in the sediment are close to zero.

The slow increase of atmospheric $CO_2$ is mainly explained by the long-term storage of material in the sediments combined with the riverine input of carbon and alkalinity, which subsequently determines how DIC is distributed into its three species $CO_2$, $HCO_3^-$, and $CO_3^{2-}$, from which only $CO_2$ can exchange with the atmosphere (Zeebe and Wolf-Gladrow, 2001).

**Table 3.** Carbon stocks (PgC) in the ocean–sediment system in our two simulations, averaged over the last 50 years.

| Reservoir | $R_{sedbox}$ | $R_{coupled}$ | Data |
|---|---|---|---|
| DIC | 35570 | 35420 | 37100[a] |
| DOC | 650 | 650 | 662[b] |
| POC | 2 | 2 | 3[a] |
| Sediment POC | < 1 | 130 | 150[c] |
| Sediment calcite | < 1 | 1060 | 800[d] |
| Sediment total | < 1 | 1390 | |

[a] Ciais et al. (2013), pre-industrial estimate
[b] Hansell et al. (2009)
[c] Emerson and Hedges (1988)
[d] Archer (1996)

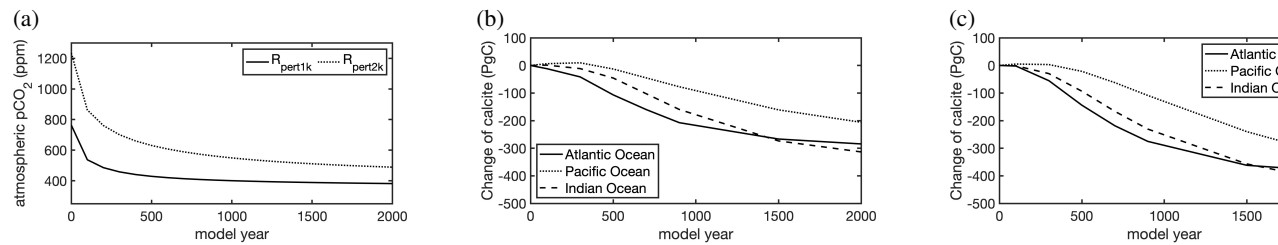

**Figure 11.** Temporal evolution of the atmospheric pCO$_2$ in the experiments with an addition of 1000 (R$_{pert1k}$) and 2000 PgC (R$_{pert2k}$) in the atmosphere (a); and the change of calcite content in sediments relative to the state before the CO$_2$ perturbation in R$_{pert1k}$ (b) and R$_{pert2k}$ (c).

### 3.2.7 Response of the coupled ocean–sediment system to perturbations in atmospheric CO$_2$

In the beginning of the two perturbation experiments R$_{pert1k}$ and R$_{pert2k}$, 1000 and 2000 PgC were added into the atmosphere, by increasing CO$_2$ concentrations to 765 and 1235 ppm, respectively (Fig. 11a). The ocean and sediment were initialized from the final state of R$_{coupled}$.

During the first 250 years, CO$_2$ concentrations sink rapidly to $\sim 480$ and 760 ppm, accompanied by a strong increase of DIC and initially a lesser one of alkalinity in the ocean. Afterwards, a much slower decline continues to the end of 2000 years. After 1000 years, about 23% of the added CO$_2$ remains in the atmosphere in R$_{pert1k}$ and 28% in R$_{pert2k}$, consistent with the range of 15–30% reported by Archer and Brovkin (2008).

The temporal evolution of sedimentary calcite stocks in the three major ocean basins reflects the carbonate compensation
feedback simulated with the coupled ocean–sediment model (Fig. 11b and c). The sedimentary calcite dissolution spreads along the ocean conveyor belt. After the perturbation, inventories quite rapidly start to decline in the Atlantic, followed with some delay by the Indian, while they remain more or less stable in the Pacific Ocean for a few centuries, before they also start to decline there. Losses then get stronger between 250 and 1000 years after the perturbation. By the year 1000 of the simulation, the net dissolution rate has already started to subside, first in the Atlantic and somewhat more slowly in the Indian and Pacific
Oceans. At that time, the total carbonate losses from the sediments in R$_{pert1k}$ amount to 220 PgC in the Atlantic, 180 PgC in the Indian, and 90 PgC in the Pacific Ocean (total: 490 PgC); in R$_{pert2k}$ the total loss is 655 PgC with the same partitioning between the ocean basins as in R$_{pert1k}$. In both experiments, the calcite loss in the Indian Ocean becomes greater than that of the Atlantic Ocean around the year 1500. After 2000 years, the total amounts of calcite dissolved from the seafloor sediments are 800 PgC in R$_{pert1k}$ and 1120 PgC in R$_{pert2k}$. The sedimentary calcite stock in the Atlantic Ocean seems to near a minimum
after 2000 years, while those in the Indian and Pacific Oceans continue to decline.

### 4 Conclusions

This paper documented the coupling of the sediment model MEDUSA2 to the marine biogeochemical model FESOM2.1-REcoM3. The coupling was realized via file exchange, the size of the annual fluxes that exchange material between the bottom

of the ocean and the sedimentary surface, was updated every 50 years. Results from a coupled simulation in a coarse resolution were presented, while a simulation with a much simpler one-layer sediment was used as reference for comparisons.

The simulation with the coupled model reasonably well reproduced the distribution of DIC, Alk, $O_2$ and nutrients found in observational data products. Biological productivity, deposition rates of particles onto sediments and burial fluxes are comparable to estimations made for deep-sea regions (below 1 km water depth), whereas they are underestimated in shallow-water regions (shallower than 1 km) and in the eastern equatorial Pacific due to the low model resolution and some missing processes in the ecosystem model.

Nutrient supply from sediments is lower in the coupled simulation than in the simulation with the one-layer sediment, particularly for nitrogen. However, the biological pump is not significantly affected by this decrease, since it is compensated by the additional riverine input of nutrients directly into the surface ocean. Changes in these two sources of nutrients lead to small changes in distribution patterns of diatoms and small phytoplankton.

After 2000 years, atmospheric $CO_2$ approaches a stable state at the pre-industrial level in our coupled simulation. About 130 PgC is stored in sediments as POC and 1060 PgC as calcite. With a coupled ocean–sediment system the model is able to simulate the carbonate compensation feedback under moderate perturbation of $CO_2$ in the atmosphere. Although most of the conclusions here are robust, one should nevertheless be aware that the exact carbon reservoirs and rates of deposition and burial presented in this paper are still results from a transient state of the simulation: a period of 2000 years is too short for the atmosphere–ocean–sediment system to reach full equilibrium, despite the sediment being pre-charged, i.e. equilibrated with sinking fluxes from an initial ocean model run.

Our model setup which includes MEDUSA2 is being further developed for parallel processing. With that, FESOM2.1-REcoM3p-MEDUSA2 can be run in higher spatial resolution for a better representation of shelf regions. Additionally, a version which includes carbon isotopes is under development. Furthermore, REcoM3p-MEDUSA2 will be used as part of the Earth System Model AWI-ESM2 (Shi et al., 2023) to explore changes in the carbon cycle during the last glacial cycle and feedbacks in the Earth's climate system.

*Code and data availability.* The source code used here has been archived on Zenodo (Ye, 2023, https://doi.org/10.5281/zenodo.8315239).

*Author contributions.* Conceptualization: YY, CV, GM; Data curation: YY; Formal analysis: YY, CV, GM; Funding acquisition: PK, CV, GM; Investigation: YY; Methodology: YY, CV, GM; Project administration: PK, CV; Software: YY, GM, CV, ÖG, MB; Resources: YY, CV; Supervision: CV, GM, PK; Validation: YY, CV, GM; Visualization: YY; Writing – original draft: YY; Writing – review & editing: YY and all co-authors

*Competing interests.* The authors declare no competing interests.

*Acknowledgements.* We thank two anonymous reviewers and the editor for their very constructive comments, which helped to improve the manuscript a lot. YY, PK, CV and MB are supported by the German Federal Ministry of Education and Research (BMBF), as Research

for Sustainability initiative (FONA); www.fona.de through the PalMod project (grant number: 01LP1919A). YY is additionally supported by the German Research Foundation (DFG) project (grant number: YE 170/2-1). MB is additionally funded through DFG-ANR project MARCARA. Financial support for GM's work on MEDUSA was provided by the Belgian Fund for Scientific Research – F.R.S.-FNRS (project SERENATA, grant no. CDR J.0123.19). GM is a Research Associate with the Belgian Fund for Scientific Research – F.R.S.-FNRS. JH and ÖG were funded by the Initiative and Networking Fund of the Helmholtz Association (Helmholtz Young Investigator Group Marine

Carbon and Ecosystem Feedbacks in the Earth System [MarESys], grant number VH-NG-1301).

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

# Appendix A: Figures

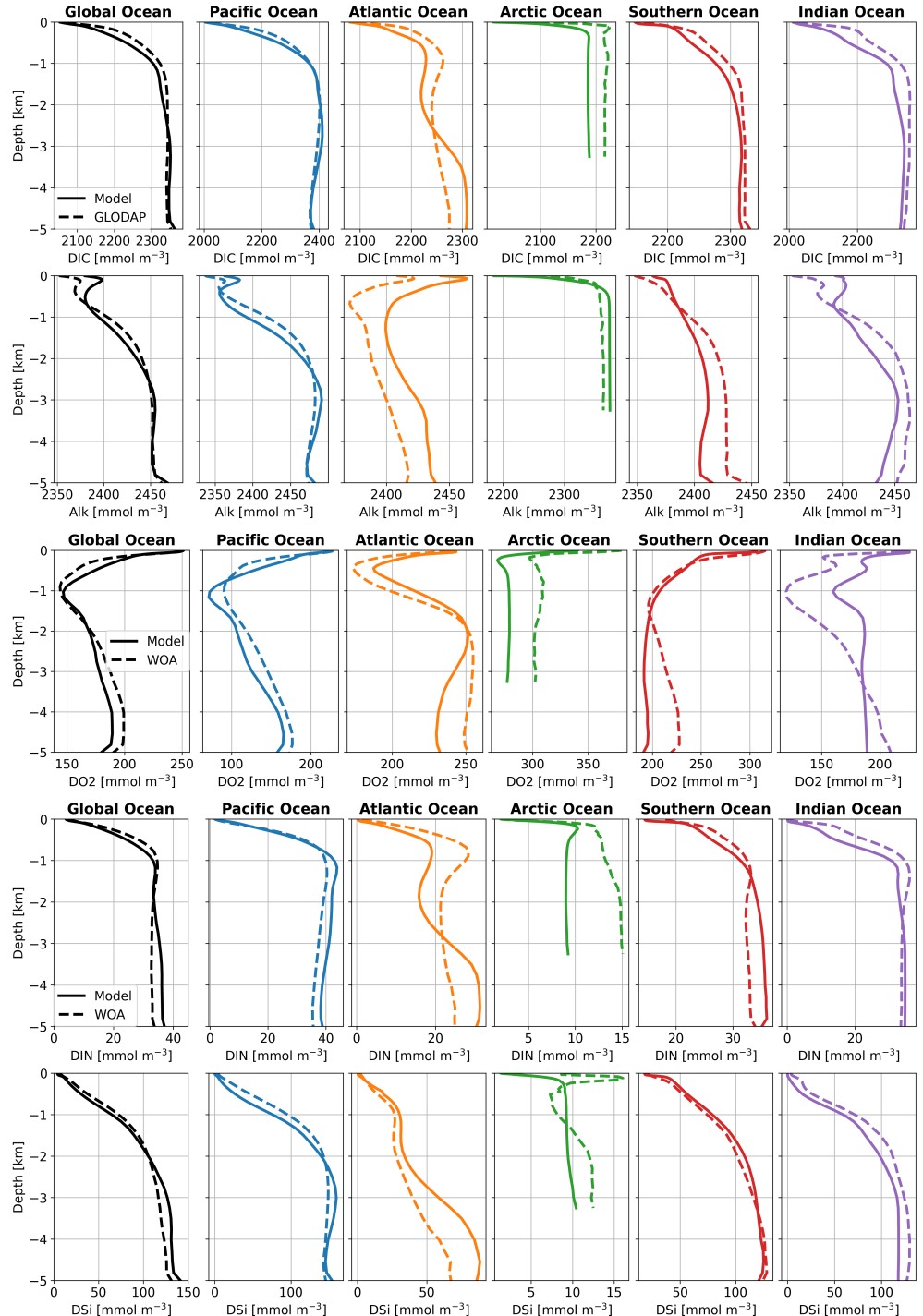

**Figure A1.** Averaged vertical profiles of DIC, Alk, $O_2$, DIN and DSi in ocean basins $(\mathrm{mmol\,m^{-3}})$ in $R_{\mathrm{spinup}}$, compared with GLODAPv2 (Large and Yeager, 2009) and WOA data (Garcia et al., 2019) which were used as initial conditions in simulations in this study.

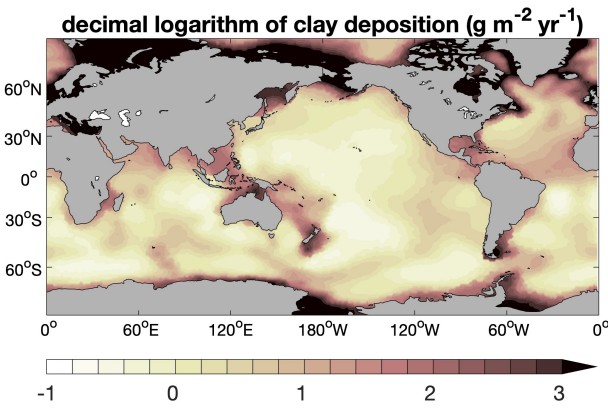

**Figure B1.** Decimal logarithm of clay flux at sediment–water interface $(\mathrm{g\,m^{-2}\,yr^{-1}})$.

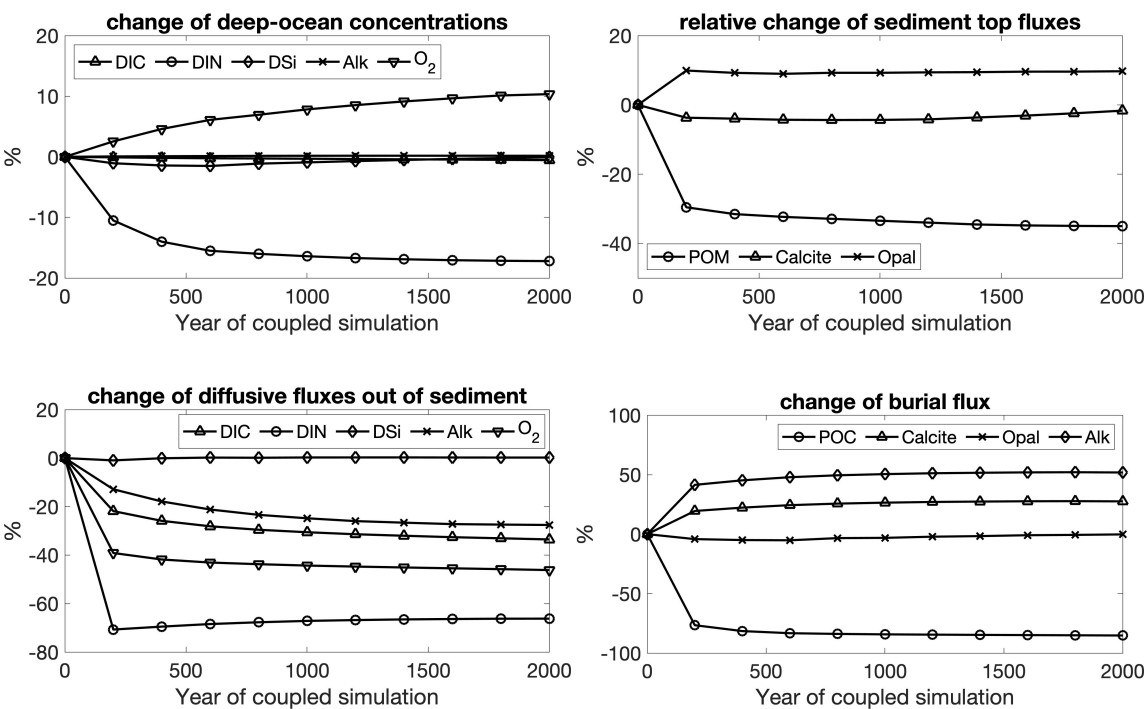

**Figure C1.** Temporal changes of deep ocean concentrations (DIC, DIN, DSi, Alk and $O_2$) (upper left) and their diffusive fluxes out of sediments (lower left); deposition rates of POM, calcite and opal onto sediment top (upper right) and their burial fluxes (lower right) during the 2000-year coupled simulation $R_{coupled}$. Changes are in percentages relative to the values at the beginning of $R_{coupled}$.

# Appendix D: FESOM2.1-REcoM3p parameters

**Table D1.** Parameters in REcoM3 modified in this study compared to Gürses et al. (2023). $Vcalc$ replaces 0.0288 in Eq. A29 in Gürses et al. (2023).

| Parameter | Value | Description | Unit |
|---|---|---|---|
| $Vdet_a$ | 0.036 | Slope of depth-dependent sinking velocity of detritus | $[\mathrm{d}^{-1}]$ |
| $Vcalc$ | 0.0072 | Slope of depth-dependent dissolution rate of calcite | $[\mathrm{d}^{-1}]$ |