# Peer review of "FESOM2.1-REcoM3-MEDUSA2: an ocean-sea ice-biogeochemistry model coupled to a sediment model"

_Geoscientific Model Development, 2023_

## Author Comment (AC1)

**Reply to the reviews of the manuscript "FESOM2.1-REcoM3-MEDUSA2: an ocean-sea ice-biogeochemistry model coupled to a sediment model"**

We much appreciate the reviewers' detailed comments, constructive suggestions as well as the critical points. Below we list all the changes that will be made in the revised manuscript and our answers to reviewers' questions, following the order of comments in the reviews.

In the following, reviewer comments start with a **C:** and are set in italics, while our responses start with a **R:** .

**Review #1:**

**C:** *Ye et al. present in their paper the coupling of the process-based Model of Early Diagenesis in the Upper Sediment (MEDUSA v.2) to an ocean biogeochemistry model (FESOM v.2.1 with reduced model resolution + REcoM v.3 in reduced complexity) as well as first results from an pre-industrial simulation with prognostic atmospheric CO2.*

*The paper is clearly written and the coupling of a process-based sediment model to an Earth system model is outlined nicely and is an important step, especially when investigating the long-term carbon cycle.*

*Therefore, I support publication in GMD and hope the authors will find my few comments below helpful and consider their implementation.*

**R:** We thank reviewer #1 for his/her support.

**General comments:**

**C:** *While I understand that the coupling of model components and work on model code in general can be very time consuming and that the focus of this study is the documentation of this coupling, I still think it would be nice to see a little more results.*

*The authors state that they use a reduced complexity version of REcoM3, REcoM3p, that is targeted for paleo simulations. Also, ocean-sediment interactions become especially interesting when looking at long timescales, such as in paleo-simulations.*

*Therefore, I think it could enrich the manuscript to see some snapshots of the coupled model under, e.g., LGM conditions and update Tables 2 and 3 with the corresponding LGM values.*

*In such an exercise, carbon isotopes would be of interest as well...*

**R:** The suggested simulations (LGM snapshots, carbon isotopes in sediments) are indeed also of interest to us and are in our research focus. However, both suggestions would require substantial additional model runs and tuning before being ready for publication. Furthermore, both suggested additional studies warrant to be analysed in more detail than can be done in a model description paper. We have chosen to submit the model description of the coupled model setup here to GMD in order to be able to have these other papers written up with less focus on the technical details and more on the scientific issues.

**C:** *Further, during comparison with observational data, the coarse resolution of the PI mesh is mentioned as a limiting factor (for example l. 246-249, 280-289) and in the conclusions an outlook*

*is given to stay tuned for not only carbon isotopes but also higher spatial resolution. In my eyes, the manuscript could further benefit from including some results of this ongoing effort, if possible, and not save it all for future publications.*

**R:** In response to the reviewer's comment, we also analyzed a simulation at a higher spatial resolution (126858 surface nodes, as in Gürses et al. (2023) and found that the POC sinking flux in shelf regions is a larger fraction (80%) of the global POC flux (close to the estimated 67-82% Muller-Karger et al. (2005); Burdige (2007)) than in the coarse-resolution (3140 surface nodes) runs presented in the manuscript, where this represented only 33% of the POC sinking flux. A paragraph about the comparison between runs in the coarse and fine resolution will be added in the model-data comparison and some results from the high-resolution run can already be found in Tab. 1. As said in response to the previous comment, the results with carbon isotopes are in our view worth a separate paper, and could not be discussed here adequately.

**Specific comments:**

**C:** *You could consider illustrating the carbonate chemistry by providing the governing equations for a better overview in the introduction.*

**R:** We thank the reviewer for the suggestion and will improve the introduction by adding equations.

**C:** *Figure 1: maybe add riverine + dust inputs to figure to close the loop?*

**R:** These fluxes will be added.

**C:** *Section 2.4.4 is not fully clear to me: Does the reported performance apply to REcoM3 or REcoM3p and to FESOM2.1 with reduced model resolution?*

**R:** It applies to FESOM2.1-REcoM3p-MEDUSA with the reduced resolution which is presented in the manuscript. This will be stated more clearly in the revised version.

**C:** *Section 3.1: you write that global vertical profiles in the model agree 'rather well' with observations from GLODAPv2. Could you give metrics? Are there differences between basins? Maybe add profiles for the different basins? Maybe also add section plots (model, obs, difference) for the main ocean basins.*

**R:** This is a good point. Here we already show the zonal distribution of DIC, Alk, DIN and $O_2$ in the Atlantic and Pacific ocean basins (Fig. 1). More figures (including the basin-averaged profiles) and details of comparison with observations will be added in the revised version.

**C:** *Mass conversation in the coupled model: It seems to me that this definitely needs to be addressed for longer (paleo-)simulations!*

**R:** Yes, we totally agree with the reviewer that ensuring mass conservation is critical for longer paleo simulations. In the submitted version we mentioned that the increase of total Si in $R_{coupled}$ by 0.8% during the 1500 model years (i.e., 0.53% kyr$^{-1}$) is very likely caused by the temporal shift in fluxes during the asynchronous coupling procedure. We now further investigated the role of the coupling frequency for mass conservation by running a coupled simulation with a higher coupling frequency (every 10 years). Although only 300 years were finished before the submission of this reply, we observed an improvement in mass conservation: Si increases by 0.40% kyr$^{-1}$ during

[Figure]

Figure 1: Zonal averaged distribution of DIC, Alk, DIN and $O_2$ in the Atlantic and Pacific Ocean (mmol m$^{-3}$).

the simulation with the higher coupling frequency. Applying a high coupling frequency for long-term simulations, however, would require much higher computation time, making long experiments unfeasible. Therefore, we will calculate a correction factor of mass conservation based on the change of Si and do the correction to the bottom water concentrations. We will test this solution during revision and add a paragraph describing this solution and show results in the revised version.

**Technical corrections:**

**C:** *l. 50: More complex scheme → More complex schemes*

**C:** *l. 111: 2) selecting of processes → 2) the selection of processes*

**C:** *l. 112: 3) writing the resulting → 3) the writing of the resulting*

**C:** *l. 127: were than partitioned → were then partitioned*

**C:** *l. 267: are reproduced in the model → are reproduced to some extent in the model?*

**C:** *l. 278: The opal belt in the equatorial eastern Pacific is smaller and less pronounced in the model than observed → not visible to me*

**R:** That is true. We apologize that the pattern was changed by changing the color bar and we did not pay attention to this detail. This figure has been redrawn with a color bar adapted to make the low-concentration areas visible. (Fig. 2).

[Figure]

Figure 2: Opal content in sediment (weight %) in R$_{medinit}$ in the submitted version.

**C:** *Caption figure 3: Horizontal averages of → Global horizontal averages of ?*

**C:** *Figure 6 and 7: label 0 seems misaligned in colorbar*

**C:** *Caption table 2: Note that the units here are Tmol year-1, not Pg year-1 → Note the different units in the table*

**C:** *Table 2: add observational estimates where available*

**R:** Thank you for the corrections - all technical corrections will be implemented.

**Review #2:**

**Summary:**

**C:** *Ying Ye and colleagues report on the coupling of the early diagenetic model MEDUSA2 with the ocean biogeochemistry model FESOM2.1-REcoM3. In order to be able to spin-up the model for multiple millennia (i.e., until the sediment-water interface (SWI) in the deep ocean is in steady-state), the authors use a lower horizontal resolution and reduce the complexity of the marine ecosystem model (i.e., simply using one generic zooplankton and detritus class instead of two for each) compared to a very recent model development paper (Gürses et al., 2023). Currently, most global Earth system models poorly represent the coupling between ocean and sediment biogeochemistry. Because the presented setup explicitly addresses the coupling of these domains, it can potentially be a very useful tool – especially for paleo-applications and simulations studying climate and marine biogeochemical feedbacks over multiple thousands of years.*

*While the model coupling itself represents a substantial contribution to Earth system modelling, the manuscript, unfortunately, lacks a proper evaluation of the performance of the new model setup and has several other weaknesses, omissions, and confusing parts. Not much new model development has been done for the manuscript – as the authors report on the coupling of two existing models. This would be okay if extensive experiments of the new coupled model evaluate its performance properly and show the added value of the new setup. Unfortunately, neither is done here. The authors only perform and show two experiments: one with the previous one-box sediment representation and one where they couple MEDUSA2. Both experiments are run under pre-industrial pCO2 for 2500 years (the new coupled configuration is only run for 1500 years from the previous model setup). Then, the authors compare some features at the end of both runs (Fig. 6, 7). The lesson learned from the results – apart from that patterns and values are slightly different - is unclear to me (e.g., is it a crucial improvement?).*

*Therefore, I cannot support the publication of this manuscript in Geoscientific Model Development. I hope my general comments will help to improve the useful coupling exercise and the evaluation of the new configuration. I suggest reconsidering the manuscript after major revision.*

**R:** We thank the reviewer for her/his comments, particularly those concerning critical points. We have considered these inputs to improve the manuscript and summarize below the changes that have been done or will be done for the revised version.

Changes that have already been done:

1. analysis of the impact of model resolution on the ratio between the sinking fluxes onto the shallow vs. the deep-ocean sediment;

2. corrected and more model-data comparisons of the sinking fluxes onto the top sediment and the burial fluxes (Tab. 1);

3. analysis of the spatial pattern and temporal development of the deep-ocean concentrations, sinking fluxes onto sediment top and diffusive fluxes out of sediment (Fig. 3);

4. analysis of the contribution of OM degradation by $O_2$ and $NO_3^-$ (Fig. 4);

5. experiments that vary the rate of oxic degradation for the OM class with a lower C:N ratio (Fig. 7);

6. a new experiment with increased coupling frequency to investigate the role of the coupling frequency for mass conservation.

More details of the changes already done are given to each of the reviewer's corresponding comments in Section "General comments".

Changes that will be done in the revised manuscript:

1. rerun the $R_{coupled}$ with the tuned degradation rate of the low C:N OM;

2. list parameters of FESOM2.1-REcoM3p and MEDUSA2 in tables;

3. complete model-data comparisons of DIC, Alk and nutrients (basin-averaged profiles);

4. experiments with mass correction;

5. all the detailed technical corrections.

**General comments:**

**C:** *The new coupled model is not thoroughly evaluated. I understand that the lower FESOM res-olution and the reduced ecosystem complexity in RecoM represent an entirely new configuration. Therefore, a more in-depth evaluation of the model than comparing it simply to global depth profiles of DIC, DIN, Dsi, and ALK (Fig. 3 – the Fig. caption states these are horizontal averages?) and showing a time-series of atmospheric pCO2 (Fig. 4) is necessary. It is not too surprising that the model gives a good match to the globally averaged GLODAP data, considering that it is initialized with it and the model is only run for 2500 years. It is necessary to show that the model – particu-larly seafloor conditions and the sediment-water interface (SWI) fluxes – is properly spun up. To evaluate if the model is in steady-state, I suggest including time-series plots of global properties such as global mean ocean O2, nutrients, DIC, DSi, ALK; export production, settling and burial fluxes of POC, CaCO3, opal, SWI fluxes of dissolved O2 and nutrients – potentially also concentrations of dissolved species at the seafloor. Maps of NPP and basin-averaged meridional-depth distributions of DIN, O2, and ALK (compared to observations) would further increase the credibility of the model.*

**R:** We agree with the reviewer that the model-data comparison could contain more details and be more extensive. Maps of NPP were shown in Fig. 6 in the submitted version and a comparison to observations will be added in the revised version, along with basin-averaged meridional-depth distributions of DIC, Alk, $O_2$ and nutrients (Fig. 1).

Fig. 3 in the submitted version shows a comparison of $R_{sedbox}$ with observations. In this model setup, the one-layer sediment was used, not the coupled sediment model MEDUSA2. In the one-layer sediment particles are fully remineralised and dissolved, and no accumulation or burial is considered. In such an ocean-sediment system, a steady state can be reached within a much shorter time than in a fully coupled ocean-sediment system. Based on our experience with the model behavior, the deep-ocean distribution of DIC, Alk and nutrients reach their steady state concentrations after 800 model years and processes mainly happening in the surface ocean, such as NPP, rather do adjust at a time scale of decades.

The reviewer is right about that 2500 years would be too short for reaching a steady state when talking about a fully coupled ocean-sediment system, since equilibration of the sediment takes place at a much longer time scale. To deal with this issue, we ran MEDUSA first for 100 kyears to fill

up the sediment ($R_{medinit}$) and reach a steady state at the ocean-sediment interface in equilibrium with the given sinking flux from $R_{sedbox}$. This state of sediments was used as the initial state for the coupled simulation $R_{coupled}$. We show the temporal changes of deep-ocean concentrations, sinking fluxes onto sediment top, diffusive fluxes out of sediments and burial during the coupled simulation in Fig. 3. One can clearly see that deep-ocean concentrations and SWI fluxes are not in a steady state but become slowly stabilised, except for the deep-ocean concentration of DIN which sinks further with time. The feature of nitrogen is probably related to the N loss through denitrification in sediments which will be explained below in our reply to a related comment. During the revision we will run $R_{coupled}$ (or rerun it with the tuned parameters) for a longer time to see how the temporal trend evolves.

Back to the reviewer's comment to the model-data comparison, here we wanted to compare $R_{sedbox}$ to observations and show that the model is validated and produces reasonable sinking fluxes and bottom water properties which are needed for the initial MEDUSA run $R_{medinit}$. In the revised version we will underline that this comparison is based on results from $R_{sedbox}$ without a fully coupled sediment and state clearly that it needs much longer to reach a steady state in a fully coupled ocean-sediment system.

[Figure]

Figure 3: Temporal changes of deep ocean concentrations (DIC, DIN, DSi, Alk and $O_2$) (upper left), sinking fluxes of POM, calcite and opal onto sediment top (upper right), diffusive fluxes of solutes out of sediments (lower left) and burial fluxes (lower right) during the 1500-year coupled simulation $R_{coupled}$. Changes are in percentages relative to the values at the beginning of $R_{coupled}$.

**C:** *The manuscript does not show and/or make use of the features that are added by coupling FESOM-REcoM to MEDUSA. MEDUSA2 simulates OM degradation with O2 and NO3. It would be interesting to see a map of the fraction of aerobic OM degradation vs denitrification. This should show a clear difference between the deeper ocean and shallower sediments with more OM input.*

**R:** This is a very helpful suggestion. Since our model setup only considers these two pathways of degradation, we calculated the fraction of oxic degradation in the total degradation, i.e., the sum of oxic degradation and denitrification. Fig. 4 (left) shows that oxic degradation contributes in large areas of the deep ocean sediments up to 100%, whereas denitrification mainly takes place in shallower sediments, which results in very low concentrations of $NO_3^-$ in the shallower sediments at high latitudes (Fig. 4, right). A chapter will be added in the revised version to discuss these model results and compare global numbers of these two processes with observations.

[Figure]

Figure 4: Fraction (%) of vertically integrated organic matter degradation by $O_2$ in total degradation (left) and the ratio of $NO_3^-$ ($R^{NO_3^-}$) concentration in pore water at the bottom of the bioturbated sedimentary mixed-layer (10 cm below the SWI) to its half saturation concentration (5 mmol m$^{-3}$).

**C:** *As mentioned in my first comment, I would like to see maps of simulated SWI-fluxes (e.g., of O2, NO3, DIC, ALK). [2.1. As a side-note: What is done in MEDUSA2 when NO3 is exhausted?]*

**R:** Here we show the diffusive fluxes of DIC, Alk, DIN and $O_2$ in $R_{coupled}$ (Fig. 5) which will be added into the revised version.

With regard to the question 'What is done in MEDUSA2 when NO3 is exhausted?': organic matter that reaches sediment depth devoid of $NO_3^-$ is simply preserved and buried. This happens in some high-latitude shallower sediments where both $NO_3^-$ concentration in pore water (Fig. 4, right) and denitrification rate are extremely low.

Beyond nitrate, MEDUSA2 has previously been used with coupled manganese and iron reduction and oxidation cycles in the sediment (Munhoven, 2021). Sulfate reduction could be easily added as a further oxidative pathway for organic matter. This was, however, not considered for the coupled simulation in this study, as including these processes, which mainly occur in shallow shelf regions with enough organic matter input, requires a much finer vertical resolution in the sediment model: the application with Mn and Fe redox processes included that was presented in Munhoven

(2021) called upon a vertical grid of 340 nodes, compared to 21 here. Here our focus is mainly on how the coupling to a sediment model influences the simulated ocean carbon cycle. Therefore, the sedimentary input to the ocean is of interest. Since our model setup in the coarse resolution does not really resolve continental shelves shallower than 150 m where sulfate, manganese and iron reduction mainly take place (Jørgensen et al., 2019; Thullner et al., 2009), the global impact of these reactions on fluxes from sediments to the ocean would be negligibly small in our simulations. We agree that these reactions should be considered if interested in processes in sediments and using a much finer model resolution.

[Figure]

Figure 5: Diffusive flux of DIC, Alk, DIN and $O_2$ from sediment to ocean (mol m$^{-2}$ day$^{-1}$). Sources for the ocean are shown as positive values.

**C:** *As described in section 2.1.2, MESUSA2 not only simulates a reactive surface sediment layer but also a core layer that records synthetic sediment cores which is a fantastic feature for paleo-applications. It would be very informative to show simulated sediment core layers for different ocean depths (e.g., a shelve vs deep ocean core) for instance during a carbon perturbation experiment.*

**R:** It is also our interest to look at the simulated sediment core layers. As the reviewer states, this becomes interesting in non-steady-state simulations. Our main motivation to couple the ocean biogeochemistry model to a sediment model is to study transient changes in carbon reservoirs over long time scales, such as the last glacial termination. This is being investigated in the coupled Earth System Model (AWI-ESM) which uses the same ocean circulation (FESOM) and biogeochemistry (REcoM) model, as mentioned at the end of the manuscript. There we run the model with a higher

resolution to also better resolve processes in shelf regions. A drawback is that more computing time is required. To this end, however, we first need to show that the model is able to produce a reasonable pre-industrial steady state, which is what we do in this paper. In this steady situation, the simulated cores are simply not interesting enough to be included in this paper. Furthermore, results of these simulations will be too comprehensive to be included in this model description paper. Therefore, we here focus on describing the technical development of the models as a first step and then, in a second step, build the paleo-applications on this basis.

**C:** *Related to the previous point: The authors want to make REcoM3 'fit for paleo-applications (see 2.2.2). Carbon isotopes are of particular interest for paleo-applications and, in my (and also the other reviewer's) opinion, should be included in this manuscript and not in a (very short) additional publication of pretty much the same authors (Butzin et al., EGUsphere). The reduced complexity configuration of the model here is particularly useful as long spin-ups are necessary to reach an equilibrium for the isotope system. I understand that carbon isotopes are currently being developed in MEDUSA2 and this might not be straight-forward in a vertically resolved diagenetic model. If this feature is not yet available, the sediment coupling of C-isotopes could for instance be simply realised by assuming no fractionation during OM remineralisation etc. in the sediment and calculating: $DIC\_13C\_swiflux = DIC\_flux\_OUT/POC\_flux\_IN * POC\_13C\_IN$*

**R:** We share the opinion of both reviewers that including coupled model setups and results for carbon isotopes is an important step to apply the model to paleo-situations. The aim of this model description paper is, however, to mainly focus on the coupling technically and the basic model performance (simulation of PI climate conditions). Therefore, we considered a coarse resolution which is commonly used for technical tests, allowing us to run a reasonable number of tuning experiments within a realistic time frame. The configuration described in the manuscript requires 2-3 weeks of computing time for 1000 model years. Including carbon isotopic traces would nearly double the computing time required of the ocean part alone. For the sediment part, including $^{13}C$ alone for all the solid C-bearing components and considering a bulk $DI^{13}C$ tracer only would require $\sim 1.8$ times as much time as the current configuration. Furthermore, Butzin et al. (2023) do not just focus on $^{13}C$ but also on $^{14}C$. For a merge of these two publications, $^{14}C$ in sediments would have to be considered as well: adding $^{14}C$ on top of $^{13}C$ to would multiply its computing time by an another factor of $\sim 1.6$, notwithstanding the order(s) of magnitude longer equilibration times required for isotopes than for normal tracers. Rerunning the coupled model with carbon isotopes to equilibrium would therefore require additional work of several months at least. As already mentioned in the reply to Reviewer #1, paleo-applications of this model will be published separately based on results from the coupled Earth System Model AWI-ESM, with a stronger focus on scientific questions than technical developments and model validation.

**C:** *The previous one-layer sediment model box, Rsedbox, is unclear to me. So Rsedbox is not really a reflective boundary but it is also not really a sediment model either – hence, there is no benthic preservation simulated with Rsedbox (lines 59-65). It would be good to clarify how the sediment box calculates the return fluxes differently compared to a reflective boundary condition. I just saw that this is described in the appendix of Gürses et al., (2023) but it would be good to also include it in this manuscript as it is necessary to understand Rsedbox.*

**R:** $R_{sedbox}$ was run with a sediment layer with a thickness of 10 cm which is not further vertically resolved. Particles sinking out of the bottom water boxes enter this sediment layer and go through

remineralization (organic particles) and dissolution (calcite and opal). The fluxes of dissolved parts back to the bottom water boxes are regulated by remineralization and dissolution rates. However, $O_2$ is not considered in the sediment layer and thus not consumed by POM remineralisaton and calcite dissolution is independent on the saturation state of the waters. Due to the remineralisation and dissolution rate, not all of the material instantaneously comes back into waters above and a small part of solids stays in the sediment layer before getting remineralized or dissolved. It has been shown in Kriest and Oschlies (2013) that this type of bottom boundary condition improves carbon and nutrient fields, compared to the slightly more simple reflective boundary condition. This sediment layer is part of FESOM2.1-REcoM3 and described briefly in Gürses et al. (2023). We will add more details about the differences between that sediment layer and MEDUSA in the revised version of the manuscript, including the different parameterisations of POM degradation and calcite dissolution.

**C:** *Related to the previous comment – Section 3.1 and table 1 is confusing. The fluxes given in Table 1 are very confusing – it is not 100% clear if these are settling or burial fluxes. The title of Tabel 1 says sinking fluxes (so settling fluxes onto the seafloor?) but the text refers to "calcite burial fluxes (line 249, 252; but then I thought Rsedbox does not simulate any preservation?). So I suppose the model estimates are settling fluxes. But some of the observational estimates are clearly burial estimates (e.g., CaCO3 data estimates are burial fluxes, as stated in Cartapanis et al., 2018). Also POC data estimates stated (50 – 2600 PgC kyr-1) probably refer to burial rates and are confusing. First, where does the 50 actually come from? I know Cartapanis et al. state it but Burdige (2007) gives 160 PgC kyr-1 as the lowest value – and these are POC burial fluxes in these publications. Often cited POC settling rates are 2628 PgC kyr-1 (Burdige, 2007), 2290 PgC kyr-1 (Dunne et al., 2007), or 930 PgC kyr-1 (Muller-Karger et al., 2004). So if the model estimates are really settling fluxes, as suggested by the title of the table and the text (see, e.g., line 240), then these values are too low and not well distributed between shallow and deeper ocean. I find the argument that the coarse resolution is responsible not convincing. Previous models with similar or even coarser resolution are able to simulate POC (and calcite) settling fluxes and preservation on the shelves much better (e.g., Palastanga et al., 2011; Hülse et al., 2018). Assuming that model estimates in Table 1 are settling fluxes – one cannot judge how well burial rates are simulated by the model. If the text is correct (in that these are calcite burial rates) then the CaCO3 burial rate in deep sea sediments ( 0.3 PgC yr-1) is 2 – 3 times larger than budget estimates (0.1 – 0.15 PgC yr-1). It would be helpful to know the mean surface sediment CaCO3 content (vs observed 34.8 wtIn summary: Please make sure the model estimates are compared to the correct observational estimates. Also, it would be very helpful to include export, settling and burial fluxes for POC, calcite and opal in the table. And also please distinguish between settling and burial fluxes for sediments shallower and deeper than 1000m, as done for the calcite data estimates in the table. This will hopefully help to understand what causes some of the mismatches in POC preservation.*

**R:** We agree that Table 1 is misleading and greatly appreciate that the reviewer has taken time to provide numerous references to the literature with precision about the kind of fluxes for which they give estimates. For the revised version, instead of summarily reporting ranges from review papers, we have collected estimates directly from single studies and thoroughly compared the definitions of shallower and deep waters. For the comparison we adapted the definition of shallower sediments with 1000 m depth from literature and will use this differentiation for shallower and deep-ocean sediment consistently throughout the revised manuscript. The total POC sinking fluxes of

simulations presented in the submitted version are then lower than the estimated sinking fluxes and the modelled burial fluxes are within the range of estimates. For the too low sinking fluxes we retuned the model and Tab. 1 show numbers from the best tuned simulation in the low resolution and from the historical simulation in Gürses et al. (2023) which uses a much higher resolution with 126858 surface nodes ($R_{high}$). As mentioned in the reply to Reviewer #1, a clearly larger shallow to deep sediments ratio is found in the high-resolution run since the shallow-water regions are much better resolved and the ratio agrees well with estimates by Muller-Karger et al. (2005) and Burdige (2007). The total sinking flux of POC in the low-resolution run ($\approx 700 \, \mathrm{PgC \, kyr^{-1}}$) is still lower than the lower end of data-based estimates ($930 \, \mathrm{PgC \, kyr^{-1}}$), while in the high-resolution run ($\approx 840 \, \mathrm{PgC \, kyr^{-1}}$) it is closer to the estimated range. This underestimation of sinking fluxes is not surprising since FESOM2.1-REcoM3 and our simplified setup both have lower global primary production compared to observations (Gürses et al., 2023). In the revised manuscript we will show new model-data comparisons of both settling and burial fluxes for the retuned $R_{sedbox}$ and the new $R_{coupled}$ which will be rerun with tuned oxidation rates for the two classes of organic matter in MEDUSA2 (see reply below and Fig. 7).

Table 1:  Sinking fluxes onto the top of sediments and burial fluxes of POC ($PgC\,kyr^{-1}$), calcite ($PgC\,kyr^{-1}$) and opal (Pmol Si $kyr^{-1}$) in simulations and measurement-based estimates, reported for the global ocean and ocean regions deeper than 1 km. Numbers for $R_{coupled}$ will be renewed in the revised version since experiments are still running. References for observations are summarized here and will be sorted in the revised version: (Burdige, 2007; Cartapanis et al., 2018; Hayes et al., 2021; Tréguer et al., 2021; Tréguer et al., 1995; Tréguer and De La Rocha, 2013; Jahnke, 1996; Muller-Karger et al., 2005; Seiter et al., 2005; Dunne et al., 2007; Cartapanis et al., 2016; Sarmiento et al., 2002; Nelson et al., 1995; Hilton and West, 2020).

| Run | Sinking fluxes | | | | | |
| --- | --- | --- | --- | --- | --- | --- |
| | POC | | calcite | | opal | |
| | all | >1 km | all | >1 km | all | >1 km |
| $R_{sedbox}$ | 696 | 464 | 324 | 318 | 76 | 70 |
| $R_{coupled}$ | 501 | 326 | 341 | 305 | 71 | 64 |
| $R_{high}$ | 841 | 159 | 436 | 367 | 45 | 34 |
| Observations | 930–5739 | 310–1029 | | | 22–40 | 79–84 |
| | Burial fluxes | | | | | |
| | POC | | calcite | | opal | |
| | all | >1 km | all | >1 km | all | >1 km |
| $R_{coupled}$ | 197 | 93 | 116 | 95 | 11 | 8.0 |
| Observations | 160–2600 | 2–300 | 280 | 130 | 7.1 | 5.9–9.2 |

**C:** *The POC wt% and the spatial distribution look not very convincing! Large areas show POC wt% > 5 (what are the maximum, mean values in these areas?) where observations show much lower values – mainly at high latitudes. In contrast, other areas were the data shows higher POC wt%, e.g., the major eastern boundary upwelling zones and the Arabian Sea, Rmedinit does not simulate any OC preservation (Fig. 5).*

**R:** Our model results indeed show a strong contrast of POC content and too high preservation in shallower sediments at high latitudes. We followed the suggestion by the reviewer and have carried out additional tuning experiments with varying the degradation rates for the two OM classes in MEDUSA2. The results are described below.

**C:** *[Also why are the observations compared to Rmedinit and not to the final results of the coupled model? I know Rmedinit is compared to Rcoupled in Fig. 8 but I don't find this comparison very informative.]*

**R:** The main difference between the two runs is that $R_{medinit}$ was run for 100 kyr to equilibrium with given boundary conditions from $R_{sedbox}$, while in $R_{coupled}$, fluxes between ocean and sediment change during the 1500 coupled years, although the changes are small due to low sedimentation rates. Therefore, with the short simulation time of 1500 years, $R_{coupled}$ does not differ much from $R_{medinit}$. This can also be seen in the diffusive fluxes out of sediments and burial fluxes ((Fig. 3), lower panel). Fig. 8 and the comparison in the submitted version should demonstrate the similarity of the two sediment simulations. Further, coupling to MEDUSA2 causes some changes in nutrient supply (Fig. 7 in the submitted version) but no substantial changes in productivity (Fig. 6 in the

submitted version). That also explains the small differences between sediment results of the two runs. The similarity and differences will be made clearer in the revised version and $R_{medinit}$ in the comparison of sediment content with observations will be replaced by $R_{coupled}$. Additionally, we will start a 5000-year MEDUSA run with constant sinking fluxes and boundary conditions from the average of the last 50 years of $R_{coupled}$ and check how sediment contents further evolve with time and discuss the results in the revised version.

**C:** *I suspect that the simplification to only use one class of detritus (line 78) in the water column might be partially responsible for the poor representation of POC wt% in the sediments (but it is impossible to be sure since no maps of POC settling fluxes are shown). The main reason however might be organic carbon degradation as simulated in MEDUSA2. MEDUSA2 simulates two classes of organic matter to approximate the different C:N stoichiometry of POC, right? What are the degradation rate constants for these classes? Is the more C-heavy class remineralized more slowly?*

*I would argue that more tuning of parameters (degradation / dissolution rate constants and/or other boundary conditions) are necessary to improve the model-data fit.*

[Figure]

Figure 6: POC sinking flux (in logarithm for easier comparison with Dunne et al. (2007)) and calcite sinking flux on sediment top in $R_{sedbox}$.

**R:** We agree with the reviewer that this point requires further analysis. As a starting point (we will add more details about this in the revised manuscript) we have produced maps of POC and calcite sinking fluxes, shown on Fig. 6. Sinking flux of POC compares well with (Dunne et al., 2007) and that of calcite also shows similar range and pattern as in (Hayes et al., 2021).

We furthermore checked the C:N ratio in sinking flux of POM (Fig. 7, upper left) and found out that C:N is generally low in shallow sediments, particularly at high latitudes, and higher in many regions in the open ocean.

The elemental composition (C:N ratio) of organic matter certainly influences its degradation time scale (Amon and Benner, 1994; Martin et al., 1987) and in the water column (FESOM2.1-REcoM3) we also considered a faster remineralisation of nitrogen compared to carbon. In MEDUSA2 we indeed applied the same oxidation rates for the two organic matter classes in the submitted manuscript. We have now started several tuning experiments where we adopted a faster degradation rate of the low-C:N organic matter class in MEDUSA2 than for the higher-C:N class. The preliminary results already show some improvements of the sedimentary POC content (see the lower panel in Fig. 7): in regions adjacent to the continents at high latitudes, POC contents still exceed 5% but the affected area is strongly reduced, while POC content in some open ocean regions decreases with the increasing degradation rate. The contrast between shallower waters and open ocean can be then mitigated. Calcite and opal content are only slightly affected by changing the degradation rate of organic matter. The upper right plot in Fig. 7 shows the original POC content in $R_{medinit}$ for comparison. This figure looks somewhat different to that in the submitted version since the color bar was adapted here to make the low-concentration areas visible. We will continue with some more tuning experiments and replace in the revised version $R_{medinit}$ and $R_{c}oupled$ with the best tuned simulations and change the model-data comparison correspondingly.

[Figure]

Figure 7: Upper panel: left: C:N ratio in POM sinking flux on sediment top in $R_{sedbox}$; right: POC in sediment (weight %) in the submitted version. Lower panel: left: POC content in the tuning runs with a 5-time enhanced degradation rate for the low C:N organic matter and right: a 10-time enhanced degradation rate.

**C:** *E.g., what about sedimentation rate: The terrestrial clay input of 2.5 E-8 mol cm-2 y-1 is spatially uniform. But should this not, as a first estimate, decrease with distance from the continents?*

*This might help with the unrealistic distribution of carbon burial between the shallow and deep ocean (as stated in table 1 and the related text).*

[Figure]

Figure 8:   Clay flux on the top of sediments.

**R:** In MEDUSA2, 'Clay' is a generic denomination for inert materials to fill the sediment. In our model setup, dust input (prescribed following Albani et al. (2014)) is primarily considered as input of lithogenic material into sediments. In addition to dust, we introduced a small uniform background flux similarly to Heinze et al. (1999) in order to represent input of lithogenic material redistributed by ocean internal processes, such as mixing and resuspension. This background flux is ten times greater than in Heinze et al. (1999) and is a result of our tuning experiments. The clay input in Fig. 8 thus derives from the sum of dust and this constant background flux. With that we partly considered a gradient decreasing with the distance from continents. We will add this figure and some explanatory sentences into the revised manuscript to describe the tuning experiments.

Rivers deliver several petagrams of suspended sediments to the ocean each year (Peucker-Ehrenbrink, 2009; Milliman and Meade, 1983). Most of these particles are deposited close the continental margins and estimates of its strength and distribution are not well constrained. Therefore, this source of lithogenic particles to marine sediments is not yet considered in our model setup. We do, however, agree that an additional riverine input of lithogenic material could reduce the too high POC fractions in shallower sediments in our simulations, and will test some assumptions in future experiments.

**C:** *The manuscript does not include any parameter values. A comprehensive table stating parameter names, values, units, and references is necessary to understand how the model is set-up. The same applies to the riverine (i.e., weathering) inputs stated in Table. Please indicate where the values come from and how they compare to observational estimates.*

**R:** A table of parameters will be added in the revised version.

**C:** *I suspect, that the loss of N (i.e., 0.8% over 1500 years of simulation) will very likely be a problem during longer model runs if not compensated for via N2 fixation and/or weathering input – especially during paleo-applications with larger contributions of denitrification. Is suggest to fix the N-leak.*

**R:** N loss of 2% $\text{kyr}^{-1}$ is mentioned in the submitted version. And we agree with the reviewer that it needs to be compensated by nitrogen fixation or weathering input. Our long-term plan is to complete the N cycle in REcoM by adding $N_2$ fixation and denitrification in the water column. For simulations in the near future, we thought of two options: the pathway via weathering would add N into the surface ocean by rivers which might have a strong effect on biological productivity, while adding N into the deep ocean (similar to the mass correction for Si mentioned above) might not significantly change the nutrient availability. We will try both options during the revision process and present results of mass correction for both Si and N in the revised version.

**C:** *Sedimentary source of iron: The text says (line 82ff.) "The sedimentary source of iron can be calculated in two ways: 1) in a fixed ratio to degradation of particulate organic nitrogen (PON) in the benthic layer as described in Gürses et al. (2023, Eq. A77 in Appendix A) or 2) in a fixed ratio to the diffusive flux of dissolved inorganic nitrogen (DIN) calculated by MEDUSA2 in coupled simulations." Can you please provide a justification for these representations and how well they approximate realistic SWI fluxes of iron. Also how well do they represent Fe fluxes under anoxic conditions – where Fe-cycling behaves very differently. This might be important depending on the paleo-applications the authors have in mind. Also, A77 does not exist in the Appendix of Gürses et al. (2023).*

**R:** Iron contained in organic particles is released into pore water during degradation of organic matter. This is the same process happening in the water column as well. Elrod et al. (2004) demonstrated a clear correlation between the iron flux out of sediments and the oxidation of organic matter on shelves, with a Fe:N ratio that is much higher than typical Fe:N ratios in sinking organic matter, implying that a large fraction of the Fe flux out of the sediment is from lithogenic material, and is mobilized by redox reactions in the sediment. Under anoxic conditions the flux of iron is increased owing to the higher solubility of ferrous iron. To represent this effect, we applied a higher Fe:N ratio (3 $\mu$molFe : 20 mmolN) for the flux of iron from the sediment to the water column than the Fe:N ratio that we used for remineralisation in the water column (1 $\mu$molFe : 30 mmolN). We will add this information into the revised version. The equation for nitrogen degradation should be A67 and will be corrected in the revised version.

**Minor comments:**

**C:** *A better motivation & explanation could be included why this model is appropriate for paleo-applications; also giving potential applications. Ideally this would be compared to existing Earth system modelling approaches, highlighting the benefits of this new model configuration.*

**R:** This is a good point. We will revised the introduction by highlighting 1) the flexible stoichiometry in REcoM3 and thus a more realistic presentation of growth limitation and marine biological carbon pump which is important for determining the sensitivity of the carbon cycle to changing climate conditions; and 2) the sediment module MEDUSA which enables simulation of the key archive of marine proxies.

**C:** *Table 2: Why does seafloor deposition (POC + CaCO3) not equal diffusive C flux out of sediment for Rsedbox?*

**R:** If we understand the reviewer correctly, the difference between 75.4 and 76.4 Tmol $\text{year}^{-1}$ was asked. This is explained by the net outgassing of 0.9 Tmol $\text{year}^{-1}$ (in the same table) which is about 0.01 PgC $\text{year}^{-1}$. That means the whole system is very close to but not completely in a

steady state.

**C:** *Section 3.3.3: Unclear where the 402 PgC come from an how it compares to the 21 PgC in table 3. Unclear what should be learned from the last bit of the section, i.e., the discussion of the carbon, alkalinity and silicon inventories not being in steady-state in the coupled run.*

**R:** We thank the reviewer for pointing this out. The discussion in the submitted version is very short and might be unclear. The 21 PgC is carbon stored as POC in the reactive layer of sediments and the 402 PgC (i.e., 381 PgC + 21 PgC) is the sum of calcite and organic carbon. This 402 PgC has two sources: 1) sediments are filled during the 100 kyears of $R_{medinit}$ and 2) due to the accumulation and burial in sediments, less carbon is released back to the ocean which can be seen by comparing the diffusive and burial fluxes in $R_{sedbox}$ and $R_{coupled}$ (Tab. 2 in the submitted version). Therefore, the carbon storage is more shifted from the atmosphere to the deep ocean and sediments, resulting a lower atmospheric $CO_2$ when coupled with MEDUSA2. A more detailed discussion will be added in the revised version.

The reason that we discussed the mass conservation of silicon in this manuscript is that it could be violated by asynchronous coupling where flux exchange between models is temporally shifted. We want to use this as a measure of the effect of asynchronous coupling on the inventory change for other tracers (e.g., carbon, nitrogen). Since carbon and alkalinity have external sources (weathering fluxes) and are not in a closed system, their inventories also change during $R_{coupled}$. Thus, it is tricky to determine how much of changes in carbon storage in different reservoirs during $R_{coupled}$ is caused by sedimentation. In the revised version, we will analyse results with mass corrections and a longer simulation time, so the problem with changing inventory will be minimized.

**C:** *Some of the methodology is unclear – I thought "FESOM2.1 was run for 1000 years to spin-up ocean circulation." (line 201) why does the 2500 year run in Fig. 4 show again ocean circulation stabilisation?*

**R:** Our description here is indeed not entirely clear. During the first 1000 years, only FESOM2.1 (the ocean model without biogeochemistry) was spun up. The results of this run are not shown in the manuscript. After that, the model was started with the physical fields produced in the first 1000-year run and REcoM3p with its original one-layer sediment, and run for another 1000 years to spin up the biogeochemistry ($R_{sedbox}$). After these 1000 years, one simulation was continued with the same setup as $R_{sedbox}$ and another one was run with the coupled MEDUSA2. Both were run for 1500 years, so that $R_{sedbox}$ has 2500 model years in total. We will explain this better in our revised version. Fig. 4 shows the stabilisation of atmospheric $CO_2$ concentration, not ocean circulation. The air-sea gas exchange is affected by the DIC concentration in the surface ocean which is regulated by the marine carbon pumps (described in the introduction). After the spin-up, the whole system is approaching a steady state and the atmospheric $CO_2$ (290.5 ppm) comes also close to its steady-state concentration (293 ppm). From 1000 to 2500 years, $R_{sedbox}$ further reaches its steady state and $R_{coupled}$ undergoes a perturbation through storing carbon in sediments and lowering sedimentary input of carbon to the ocean. Therefore, more $CO_2$ is taken up by the ocean and the atmospheric $CO_2$ decreases, until the gas exchange is balanced after several hundreds to about 1000 years. During this period, ocean circulation remains stabilised and only the distribution of carbon is changed from one to another state.

**C:** *It is also confusing that two experiments are named Rsedbox – one described in 2.4.1 and then*

*one described in 2.4.3 (and shown in Fig. 4?) The description of Rcoupled is also not 100%
clear. The text states: "(2) a coupled simulation Rcoupled was conducted for 1500 model years first
using the output from Rmedinit as sedimentary input of DIC, Alk and nutrients." (lines 231-214)
I suppose this means, you start with the SWI-exchanges calculated in Rmedinit (i.e., for the first
50 years), after that MEDUSA is called every 50 years.*

**R:** Yes, it is correct. As explained in our reply to the previous comment, $R_{sedbox}$ has actually run
for 2500 years in total, where the first 1000 years come from the biogeochemistry spin-up stage.
That seafloor deposition fluxes and the bottom water concentrations obtained at the end of the
biogeochemistry spin-up were used for the first MEDUSA2 run $R_{medinit}$ (whose purpose was to
pre-fill MEDUSA2's sediment in an approximate steady-state way); $R_{sedbox}$ was continued further
1500 years to produce results that are comparable (i. e., that cover the same simulation length) to
those obtained with MEDUSA2 coupled to FESOM2.1-REcoM3 ($R_{coupled}$). We will rephrase the
description of experiments and clarify this in the revised version.

**C:** *Lines 268 ff. What does this refer to? How is this different to Fig. 5?*

**R:** The model-data comparison of sinking fluxes was not shown in the submitted version. They
are added here (Fig. 6) and will be shown in the revised version as well.

**References**

S. Albani, N. M. Mahowald, A. T. Perry, R. A. Scanza, C. S. Zender, N. G. Heavens, V. Maggi,
J. F. Kok, and B. L. Otto-Bliesner. Improved dust representation in the Community Atmosphere
Model. Journal of Advances in Modeling Earth Systems, 6(3):541–570, September 2014. ISSN
19422466. doi: 10.1002/2013MS000279.

R. Amon and R. Benner. Rapid cycling of high-molecular-weight dissolved organic matter in the
ocean. Nature, 369:549–552, 1994. doi: 10.1038/369549a0.

David J. Burdige. Preservation of organic matter in marine sediments: Controls, mechanisms, and
an imbalance in sediment organic carbon budgets? Chemical Reviews, 107(2):467–485, February
2007. ISSN 0009-2665, 1520-6890. doi: 10.1021/cr050347q.

M. Butzin, Y. Ye, C. Völker, Ö. Gürses, J. Hauck, and P. Köhler. Carbon isotopes in
the marine biogeochemistry model FESOM2.1-REcoM3. EGUsphere, 2023:1–36, 2023. doi:
10.5194/egusphere-2023-1718.

Olivier Cartapanis, Daniele Bianchi, Samuel L. Jaccard, and Eric D. Galbraith. Global pulses of
organic carbon burial in deep-sea sediments during glacial maxima. Nature Communications, 7,
2016. doi: 10.5194/gmd-2023-68.

Olivier Cartapanis, Eric D. Galbraith, Daniele Bianchi, and Samuel L. Jaccard. Carbon burial in
deep-sea sediment and implications for oceanic inventories of carbon and alkalinity over the last
glacial cycle. Clim. Past, 14(11):1819–1850, November 2018. ISSN 1814-9332. doi: 10.5194/
cp-14-1819-2018.

John P. Dunne, Jorge L. Sarmiento, and Anand Gnanadesikan. A synthesis of global particle
export from the surface ocean and cycling through the ocean interior and on the seafloor. Global
Biogeochemical Cycles, 21(4):GB4006, 2007. doi: 10.1029/2006GB002907.

Virginia A. Elrod, William M. Berelson, Kenneth H. Coale, and Kenneth S. Johnson. The flux of iron from continental shelf sediments: A missing source for global budgets. Geophysical Research Letters, 31(12), 2004. doi: 10.1029/2004GL020216.

Ö. Gürses, L. Oziel, O. Karakuş, D. Sidorenko, C. Völker, Y. Ye, M. Zeising, M. Butzin, and J. Hauck. Ocean biogeochemistry in the coupled ocean–sea ice–biogeochemistry model FESOM2.1–REcoM3. Geoscientific Model Development, 16(16):4883–4936, 2023. doi: 10.5194/gmd-16-4883-2023.

Christopher T. Hayes, Kassandra M. Costa, Robert F. Anderson, Eva Calvo, Zanna Chase, Ludmila L. Demina, Jean-Claude Dutay, Christopher R. German, Lars-Eric Heimbürger-Boavida, Samuel L. Jaccard, Allison Jacobel, Karen E. Kohfeld, Marina D. Kravchishina, Jörg Lippold, Figen Mekik, Lise Missiaen, Frank J. Pavia, Adina Paytan, Rut Pedrosa-Pamies, Mariia V. Petrova, Shaily Rahman, Laura F. Robinson, Matthieu Roy-Barman, Anna Sanchez-Vidal, Alan Shiller, Alessandro Tagliabue, Allyson C. Tessin, Marco Van Hulten, and Jing Zhang. Global ocean sediment composition and burial flux in the deep sea. Global Biogeochemical Cycles, 35(4):e2020GB006769, April 2021. ISSN 0886-6236, 1944-9224. doi: 10.1029/2020GB006769.

C. Heinze, E. Maier-Reimer, A. M. E. Winguth, and D. Archer. A global oceanic sediment model for long-term climate studies. Global Biogeochemical Cycles, 13(1):221–250, March 1999. ISSN 08866236. doi: 10.1029/98GB02812.

Robert G. Hilton and A. Joshua West. Mountains, erosion and the carbon cycle. Nature Reviews Earth and Environment, 1(6), 2020. doi: 10.1038/s43017-020-0058-6.

Richard A. Jahnke. The global ocean flux of particulate organic carbon: Areal distribution and magnitude. Global Biogeochemical Cycles, 10(1):71–88, 1996. doi: 10.1029/95GB03525.

Bo Barker Jørgensen, Alyssa J. Findlay, and Andr'e Pellerin. The biogeochemical sulfur cycle of marine sediments. Frontiers in Microbiology, 10, 2019. doi: 10.3389/fmicb.2019.00849.

I. Kriest and A. Oschlies. Swept under the carpet: Organic matter burial decreases global ocean biogeochemical model sensitivity to remineralization length scale. Biogeosciences, 10(12):8401–8422, December 2013. ISSN 1726-4189. doi: 10.5194/bg-10-8401-2013.

John H. Martin, George A. Knauer, David M. Karl, and William W. Broenkow. Vertex: carbon cycling in the northeast pacific. Deep Sea Research Part A. Oceanographic Research Papers, 34(2):267–285, 1987. ISSN 0198-0149. doi: 10.1016/0198-0149(87)90086-0.

John D. Milliman and Robert H. Meade. World-wide delivery of river sediment to the oceans. JGeol, 91(1):1–21, 1983. doi: 10.1086/628741.

Frank E. Muller-Karger, Ramon Varela, Robert Thunell, Remy Luers sen, Chuanmin Hu, and John J. Walsh. The importance of continental margins in the global carbon cycle. Geophysical Research Letters, 32(1):L01602, 2005. doi: 10.1029/2004GL021346.

Guy Munhoven. Model of Early Diagenesis in the Upper Sediment with Adaptable complexity – MEDUSA (v. 2): A time-dependent biogeochemical sediment module for Earth system models, process analysis and teaching. Geoscientific Model Development, 14(6):3603–3631, June 2021. ISSN 1991-9603. doi: 10.5194/gmd-14-3603-2021.

David M. Nelson, Paul Tréguer, Mark A. Brzezinski, Aude Leynaert, and Bernard Qu'eguiner. Production and dissolution of biogenic silica in the ocean: Revised global estimates, comparison with regional data and relationship to biogenic sedimentation. Global Biogeochemical Cycles, 9 (3):359–372, 1995. doi: 10.1029/95GB01070.

Bernhard Peucker-Ehrenbrink. Land2sea database of river drainage basin sizes, annual water discharges, and suspended sediment fluxes. Geochemistry, Geophysics, Geosystems, 10(6), 2009. doi: 10.1029/2008GC002356.

J. L. Sarmiento, J. Dunne, A. Gnanadesikan, R. M. Key, K. Matsumoto, and R. Slater. A new estimate of the $CaCO_3$ to organic carbon export ratio. Global Biogeochemical Cycles, 16(4): 1107, 2002. doi: 10.1029/2002GB001919.

Katherina Seiter, Christian Hensen, and Matthias Zabel. Benthic carbon mineralization on a global scale. Global Biogeochemical Cycles, 19(1), 2005. doi: 10.1029/2004GB002225.

Martin Thullner, Andrew W. Dale, and Pierre Regnier. Global-scale quantification of mineralization pathways in marine sediments: A reaction-transport modeling approach. Geochemistry, Geophysics, Geosystems, 10(10), 2009. doi: https://doi.org/10.1029/2009GC002484.

Paul Tréguer, David M. Nelson, Aleido J. Van Bennekom, David J. DeMaster, Aude Leynaert, and Bernard Quéguiner. The silica balance in the world ocean: A reestimate. Science, 268(5209): 375–379, 1995. doi: 10.1126/science.268.5209.375.

Paul J. Tréguer and Christina L. De La Rocha. The world ocean silica cycle. Annual Review of Marine Science, 5(1):477–501, 2013. doi: 10.1146/annurev-marine-121211-172346.

Paul J. Tréguer, Jill N. Sutton, Mark Brzezinski, Matthew A. Charette, Timothy Devries, Stephanie Dutkiewicz, Claudia Ehlert, Jon Hawkings, Aude Leynaert, Su Mei Liu, Natalia Llopis Monferrer, María López-Acosta, Manuel Maldonado, Shaily Rahman, Lihua Ran, and Olivier Rouxel. Reviews and syntheses: The biogeochemical cycle of silicon in the modern ocean. Biogeosciences, 18(4):1269–1289, February 2021. ISSN 1726-4189. doi: 10.5194/bg-18-1269-2021.

---

## Author Response (AR1)

**Letter for resubmitting the manuscript "FESOM2.1-REcoM3-MEDUSA2: an ocean-sea ice-biogeochemistry model coupled to a sediment model"**

Dear Dr. Arndt and dear reviewers,

we are writing to resubmit the revised manuscript titled "FESOM2.1-REcoM3-MEDUSA2: an ocean-sea ice-biogeochemistry model coupled to a sediment model," originally submitted on September 4, 2023. We appreciate the constructive feedback provided by the reviewers and have thoroughly addressed their comments and suggestions in this revised version. This resubmission is made with the consent of all co-authors.

We would like to highlight again the key revisions made in response to the reviewers' feedback:

1. We slightly changed the structure of the manuscript by changing the names of simulations and titles of subsections so that the revised manuscript more focuses on the coupled simulation with FESOM2.1-REcoM3p-MEDUSA2;

2. MEDUSA2 was re-tuned to with a higher degradation rate for the low C:N organic matter;

3. Mass correction was applied in the coupled simulation to minimise the effect by asynchronous coupling;

4. The loss of nitrogen through denitrification in sediments was compensated by adding the corresponding amount of DIN in the bottom water boxes;

5. Based on the changes of the model code mentioned above, $R_{coupled}$ was rerun for a longer period: 2000 model years instead of 1500 years in the submitted version. The results of this new $R_{coupled}$ were shown and discussed in the revised manuscript;

6. More analyses of MEDUSA2 results were done and three new subsections focusing on the degradation of organic matter in sediments, solute exchange across the sediment-water interface and the burial fluxes were added into the result section and compared to available data where possible;

7. Two tables of model parameters were added in the appendix. For FESOM2.1-REcoM3p it shows the parameters modified in this study (for the configuration targeted for paleo-application) compared to Gurses et al. (2023). And for MEDUSA2 the table shows all reaction rate law expressions and parameter values used in this study;

8. The model-data comparisons of DIC, Alk, $O_2$ and nutrients (basin-averaged profiles) were completed to illustrate the initial conditions of the coupled simulation and the impact of the complex sediment model on marine biogeochemistry;

9. The model-data comparisons of seafloor deposition rates and burial fluxes were revised and Table 1 was corrected and completed.

While working on the revisions we decided in a few points to deviate slightly from the changes that we had announced in our response to the reviewers. This concerns the following points:

1. We gave in the Introduction a clearer description of the carbonate chemistry with a citation that describes its details and added chemical notations. We chose, however, not to show

the basic chemical equilibrium relations of the carbonate system as Reviewer #1 suggested, because these can be easily found in the cited publication;

2. We did show the zonal distribution of DIC, Alk, DIC and $O_2$ in two ocean basins in the response letters. In the revised manuscript, however, we decided to show the basin-averaged vertical profiles of DIC, Alk, DIN, DSi and $O_2$ of the simulation with a one-layer sediment representation and of the simulation with MEDUSA2, in comparison with observations instead, since we think that these illustrate the model performance in each basin in a much clearer way and allow for a better comparison with observation-based values.

We believe that the revised version addresses the concerns raised during the review process and have substantially improved the manuscript, making it more robust and suitable for publication in GMD.

Thank you for your time and consideration. We look forward to hearing from you soon.

Sincerely,

Ying Ye for all co-authors

---

## Referee Report (RR1)

**2nd Review of "FESOM2.1-REcoM3-MEDUSA2: an ocean-sea ice-biogeochemistry model coupled to a sediment model"**

by

Ying Ye, Guy Munhoven, Peter Köhler, Martin Butzin, Judith Hauck , Özgür Gürses, and Christoph Völker

**Summary:**

Ying Ye and colleagues resubmitted a revised version of their manuscript with significant changes to address the comments of both reviewers. Most notably, the authors changed the degradation rate constant for the low C:N OM class in MEDUSA2, they performed mass corrections to account for the asynchronous coupling of the models and the loss of nitrogen through denitrification in sediments, three new subsections were added to the Results (3.2.2 – 3.2.4, briefly describing MEDUSA2 results at the end of coupled simulation), and Figure 3 & Table 1 (summarizing their globally integrated results) has been revised and improved – even though problems still exist with the reported opal fluxes (see comment #1.3. I really appreciate the improved Fig. 5.

While the authors have made substantial improvements to the manuscript, there are still significant weaknesses that need to be addressed (some of which were raised in the previous reviews but have not been addressed in a satisfying manner). In my view, most importantly, it would be beneficial to include more and better "tuning" experiments (e.g., to enhance results for modern OM preservation + at least one idealized transient simulation). This will help to convince readers of the model framework's appropriateness to simulate not only appropriate steady-state modern and LGM conditions but also transient Earth system dynamics, as intended by the authors in future studies. Having a model configuration that simulates modern marine conditions well should be the minimum goal for such a *GMD* paper, especially considerinhg that this configuration will be the reference for future studies!

Therefore, I still cannot support the publication of the manuscript in *Geoscientific Model Development*. I summarize my main concerns below and I hope that my general comments will help to improve the manuscript further.

**General comments:**

Comment #1.1: Improve the simulation patterns of preservation in the sediments:

The poor model-data fit in Fig. 6 can not only be explained by the lower resolution. As stated in the previous review: "Previous models with similar or even coarser resolution are able to simulate POC (and calcite) settling fluxes and preservation on the shelves much better (e.g., Palastanga et al., 2011; Hülse et al., 2018; Ridgwell & Hargreaves, 2007)."

Taking OM as an example, I think there a multiple reasons for why the wt% patterns are not very realistic (that could/should be addressed):

1. The settling fluxes at high latitudes are (potentially) too large (comparing your Fig. 4a with Fig. 5a of Dunne et al. – note, that their color scale is very different), whereas the global total seems to be way too small (see Tab. 1). So the pelagic POC degradation should be improved first as this will influence the OM available for benthic preservation.

2. OM is not further oxidized when nitrate is exhausted – this is not correct and will cause too much OM preservation in these grid-cells: Judging by Fig. 7 (right, or Fig. 4 in replies to reviewers), NO3 is zero in a large fraction of higher latitude cells → in these grid-cells  too much OM is preserved (see Fig. 6)

3. Better tuning of the OM fractions and degradation rate constants in MEDUSA2 will also help. (For the revision the authors simoply increased one degradation rate constant by a factor of 5 and 10.)

Instead, rate constants could depend on seafloor depth, sedimentation rate or OM settling flux (see e.g., Boudreau, Springer Berlin, 1997). This would potentially not only improve the model-data fit but also responds to changing environmental conditions when simulating paleo-conditions.

Also, as the pelagic model only represents one OM fraction: How do you specify the two OM fractions in MEDUSA2 from this? I don't think this is discussed in the manuscript.

The results, of course, don't need to be perfect but should be better than presented in Fig. 6. And considering a run time of "2-3 weeks for 1000 model years" a few more experiments to improve the configuration are reasonable.

Comment #1.2:

The global burial fluxes of POC (110 PgC kyr-1) and calcite (115 PgC kyr-1) look good (see Table 1). Please check the highest POC data estimate in Tab. 1 (i.e., 2600 PgC kyr-1) is this not a settling flux?

Comment #1.3: There seems to be something wrong with the opal fluxes:

How can the opal settling flux in areas >1km (79-84 Pmol Si kyr-1) be larger than the global flux (22-40 Pmol Si kyr-1) in Table1? Also the units are different in the text (Tmol Si yr-1) and the Tab. 1 (Pmol Si kyr-1). And why is the global opal burial flux even larger (82 Pmol Si kyr-1). And compared to this the observed global burial is tiny (7.1 Pmol Si kyr-1).

Comment #2:

Figure 6: The new color-bars for the model results (right) are finer than the color-bar for the observations (left). Therefore, model and observations cannot properly be compared as more features appear in the model results than in the observations (this was introduced to address one of the comments of reviewer #1).

Comment #3:

A transient experiment or paleo-application. This would be highly informative and was suggested by both reviewers (e.g., 1st comment of Reviewer #1) but has not been addressed. At least the model could be used to simulate an idealised perturbation experiment to showcase that it can be used for transient applications and that the sediment properties respond. In particular, because this is mentioned as one of the main motivations for configuring this model.

Comment #4:

I was also not very convinced by the authors reasoning for why they did not include some output of C-isotopes. This was also suggested by both reviewers and is highly relevant for a model that will be applied to paleo-applications.

Comment #5:

Why a mass correction to account for the asynchronous coupling is necessary is unclear to me. Please explain and justify this better in the text. My understanding is, that while this might affect the inventories & fluxes during the spin-up phase the system should eventually come to a (new) steady-state.

Comment #6:

The higher spatial resolution result ($R_{high}$): I understand, that this was included to address parts of the 2nd comment of Reviewer #2. However, these results are rather out of place here. Also the authors argument that this experiment shows that the coarse model resolution of $R_{coupled}$ is mainly responsible for the low settling fluxes (lines 316 – 320) is at least insufficient: It is unclear if model resolution is

the only boundary condition/parameter that changes between both configurations. Anyhow, there are multiple ways to increase the POC settling fluxes in the lower resolution $R_{coupled}$ configuration (e.g., by increasing the low export production as acknowledged by the authors, or decreasing pelagic POC remineralization rates).

Comment #7:
Fig. 3: I suggest including the profiles at the end of $R_{coupled}$ here as well because these are the new results presented and tested in the manuscript.

**A few minor comments:**

Fig. 6: What is the simulated wt% here? Is it mean over the bioturbated layer, at 10cm, or something else? And what are the corresponding values of Hayes et al. (2021) representing?

Fig. 7: I find it not very intuitive why you show $R_{NO3}$ here. Can you please motivate this. If sulfate reduction would be included in the model (more important than Mn, or Fe-reduction on a global scale) then one could nicely show the different fractions.

Fig. C1: Why is there such a large drop in the calcite burial flux even though the calcite settling flux increases over the experiment?

---

## Author Response (AR2)

**Reply to the second review of the manuscript**
**"FESOM2.1-REcoM3-MEDUSA2: an ocean-sea ice-biogeochemistry model coupled to a sediment model"**

We appreciate the reviewer's constructive suggestions as well as the critical points. Below we list the changes made in the revised manuscript and our answers to reviewer's questions, following the order of comments in the reviews.

In the following, reviewer comments start with a **C:** and are set in italics, while our responses start with a **R:** .

**Summary:**

**C:** *Ying Ye and colleagues resubmitted a revised version of their manuscript with significant changes to address the comments of both reviewers. Most notably, the authors changed the degradation rate constant for the low C:N OM class in MEDUSA2, they performed mass corrections to account for the asynchronous coupling of the models and the loss of nitrogen through denitrification in sediments, three new subsections were added to the Results (3.2.2–3.2.4, briefly describing MEDUSA2 results at the end of coupled simulation), and Figure 3 & Table 1 (summarizing their globally integrated results) has been revised and improved – even though problems still exist with the reported opal fluxes (see comment #1.3. I really appreciate the improved Fig. 5.*

*While the authors have made substantial improvements to the manuscript, there are still significant weaknesses that need to be addressed (some of which were raised in the previous reviews but have not been addressed in a satisfying manner). In my view, most importantly, it would be beneficial to include more and better "tuning" experiments (e.g., to enhance results for modern OM preservation + at least one idealized transient simulation). This will help to convince readers of the model framework's appropriateness to simulate not only appropriate steady-state modern and LGM conditions but also transient Earth system dynamics, as intended by the authors in future studies. Having a model configuration that simulates modern marine conditions well should be the minimum goal for such a GMD paper, especially considering that this configuration will be the reference for future studies!*

*Therefore, I still cannot support the publication of the manuscript in Geoscientific Model Development. I summarize my main concerns below and I hope that my general comments will help to improve the manuscript further.*

**R:** We thank the reviewer for acknowledging the improvements of the manuscript.

One of the weaknesses of our model that the reviewer pointed out is the unsatisfying model-data fit of the sediment preservation, particularly the preservation of total particulate organic carbon (TOC) at high latitudes. During the revision we made much effort to improve the TOC preservation by:

- conducting sensitivity experiments with respect to the settling flux to the seafloor

- adding sulfate reduction as an additional organic matter (OM) degradation pathway;

- extending the sediment model to a greater depth (50 cm now, instead of 10 cm before) within

the sediment, resolved with more vertical layers (71 node-grid now, compared to 21-node grid before);

- and improving the global sedimentation rate distribution with an additional lithogenic ("clay") input, as it turned out that the dust input flux was insufficient to produce realistic mass accumulation rates at shallow seafloor depths.

These changes have clearly improved the model-data fit, especially through considering OM degradation by sulfate reduction and enhanced sedimentation rates in shallower waters. Details of model changes are given in the reply to the general comments. Based on these improvements, the coupled FESOM2.1-REcoM3-MEDUSA2 simulation ($R_{coupled}$) was rerun and all model results have been updated in the revised version.

The other major critical point that the reviewer raised was that we did not show any results of a LGM or transient simulation. In our previous response letter, we mentioned that LGM simulations have been started with the fully coupled setup–AWI-ESM2, and those results will be published in another paper focusing on mechanisms driving the glacial $CO_2$ draw-down. To demonstrate that the ocean-only setup with MEDUSA can be used to study transient climate changes, we now conducted two transient experiments which describe the reaction of the ocean–sediment system after adding 1000 and 2000 PgC into the atmosphere, respectively. With those experiments, the interactions between the atmosphere, ocean and sediment under perturbation in the atmospheric $CO_2$ can be examined. More details are provided below in the reply to the general comments. The results of those experiments are presented in the revised manuscript.

**General comments:**

**C:** *Comment #1.1: Improve the simulation patterns of preservation in the sediments: The poor model-data fit in Fig. 6 can not only be explained by the lower resolution. As stated in the previous review: "Previous models with similar or even coarser resolution are able to simulate POC (and calcite) settling fluxes and preservation on the shelves much better (e.g., Palastanga et al., 2011; Hülse et al., 2018; Ridgwell & Hargreaves, 2007)."*

*Taking OM as an example, I think there a multiple reasons for why the wt% patterns are not very realistic (that could/should be addressed):*

*1. The settling fluxes at high latitudes are (potentially) too large (comparing your Fig. 4a with Fig. 5a of Dunne et al. – note, that their color scale is very different), whereas the global total seems to be way too small (see Tab. 1). So the pelagic POC degradation should be improved first as this will influence the OM available for benthic preservation.*

**R:** We agree that the modelled TOC wt% in Fig. 6 in the submitted version does not really match the observations. The reviewer is also right that the settling flux onto the sediment surface and the OM degradation in sediments are the two factors affecting the TOC content in sediments. We first compared the export production in our simulation with observation-driven model results (Clements et al., 2023, and studies compared there) and our result is within the range of those studies and relatively close to (Dunne et al., 2007). Estimated settling fluxes on seafloor vary over at least one order of magnitude at high latitudes (Dunne et al., 2007; Hayes et al., 2021). It is thus difficult to judge if the fluxes in our model are too high.

In REcoM3, settling fluxes onto the sediment surface are determined by sinking velocity and remineralisation of POC and the latter is a function of temperature and POC concentration. Sinking velocity of POC was already tuned to match the observed basin averaged vertical profiles of DIC, Alk, $O_2$, DIN and DSi (Fig. 3 in the submitted version) which show a good agreement with observations. And further changes in the sinking velocity will not affect the contrast between high and low latitudes. Thus, our tuning work focused on the temperature dependence of remineralisation. Several sensitivity experiments were conducted where Q10 for microbial degradation of POC in the water column was lowered step-wise from 2.3 to 1.5 so that OM degradation becomes faster at low temperatures (high latitudes) and slower at high temperatures. The settling fluxes of POC show correspondingly a stronger decline in large areas at high latitudes, while they are also reduced almost in the entire global ocean (Fig. 1). The contrast of sediment carbon preservation between the high-latitude shallower waters and the large area of the global open ocean does not change as much as the settling flux and much smaller areas at high latitudes are affected. A Q10 value much lower than 1.5 would be not reasonable for known microbial activities (Laufkötter et al., 2017).

Furthermore, the good agreement of basin-wise averaged profiles of DIC, Alk, $O_2$, DIN and DSi with GLODAPv2 and WOA data also provides an evidence for a realistic water-column degradation of OM in our model. Therefore, we did not further tune the model regarding the settling fluxes but focused on the degradation processes in sediments.

[Figure]

Figure 1: Differences in the settling flux of POC onto the sediment surface ($\mathrm{mmolC\,m^{-2}\,day^{-1}}$) and TOC wt% in sediments between the sensitivity experiment with Q10=1.5 and $\mathrm{R}_{init}$ in the submitted version.

**C:** *2. OM is not further oxidized when nitrate is exhausted – this is not correct and will cause too much OM preservation in these grid-cells: Judging by Fig. 7 (right, or Fig. 4 in replies to reviewers), NO3 is zero in a large fraction of higher latitude cells → in these grid-cells too much OM is preserved (see Fig. 6)*

**R:** Too much TOC is indeed preserved at high latitudes where $NO_3$ is exhausted. Unlike the the models cited above, MEDUSA includes a consistent $NO_3$ balance. If a simple first order approach is used for anoxic degradation (as, e.g., Palastanga do), the implicit assumption is that there are always enough oxidants besides $O_2$. However, the oxidant balance is not closed, and since $NO_3$ is an oxidant and a nutrient, the nutrient cycle is unbalanced as well. Therefore, we closed the nutrient balance which results in a side-effect of excessive OM preservation in the submitted manuscript.

To better simulate OM degradation when $NO_3$ is exhausted, we now added $SO_4$ reduction into the reaction network after aerobic degradation and nitrate reduction. The same reaction rate constants

are applied for all degradation pathways, while they differ between the two OM classes over two orders of magnitude. A series of sensitivity runs were conducted to by varying the reaction rate constants to fit the observed TOC content in the surface sediment. $SO_4$ reduction substantially reduced the OM preservation at high latitudes and with the enhanced sedimentation rate in shallower waters together, the too strong contrast between the high-latitude shallower waters and the open ocean has been clearly improved as well (Fig. 2a and b).

Model details of $SO_4$ reduction and the new simulations are provided in the revised manuscript. Additionally, a detailed documentation is provided as supplementary material to describe the MEDUSA2 configuration used in the study.

[Figure]

Figure 2: TOC wt% averaged over the upper 10 cm of the surface sediment in $R_{init}$ in the submitted version (a), in the new simulations considering $SO_4$ reduction (b) and additionally with improved clay input (c), and TOC wt% in Hayes et al. (2021) (d).

**C:** *3. Better tuning of the OM fractions and degradation rate constants in MEDUSA2 will also help. (For the revision the authors simply increased one degradation rate constant by a factor of 5 and 10.) Instead, rate constants could depend on seafloor depth, sedimentation rate or OM settling flux (see e.g., Boudreau, Springer Berlin, 1997). This would potentially not only improve the model-data fit but also responds to changing environmental conditions when simulating paleo-conditions.*

**R:** Our configuration for the submitted version considered different degradation rates for the two OM classes: OM with a higher C:N ratio is 100-fold more slowly degraded through reaction with oxygen than OM with a lower C:N ratios. Degradation of different OM classes by nitrate reduction, however, had the same rate constant. In the revised version, the same rate constants are used for

oxic degradation, nitrate and sulfate reduction, but they differ between OM classes. With the additional degradation through sulfate reduction combined with the deepening of the sediment to 50 cm, we managed to well reproduce the observed pattern of TOC preservation.

**C:** *Also, as the pelagic model only represents one OM fraction: How do you specify the two OM fractions in MEDUSA2 from this? I don't think this is discussed in the manuscript.*

**R:** The ocean model REcoM3 simulates only one POM but with a flexible C:N ratio. To represent the entire range of the variable stoichiometry in the ocean model and keep the mass conserved between the ocean and sediment, we defined two POM classes in MEDUSA with the minimum and maximum C:N ratio found in settling fluxes produced by REcoM3 and partitioned the settling fluxes of PON (particulate organic nitrogen) into these two classes and then calculated the corresponding POC fluxes based on the fixed C:N ratios of the two POM classes. This was described in detail in the submitted manuscript from L170 to L182.

**C:** *The results, of course, don't need to be perfect but should be better than presented in Fig. 6. And considering a run time of "2-3 weeks for 1000 model years" a few more experiments to improve the configuration are reasonable.*

**R:** We agreed to carry out more experiments to improve the configuration, but this assessment of time that is needed was not realistic. It is true that calculating 1000 model years requires about 2 weeks. However, a lot of simulations have to be done for tuning a model and they can not always be run at the same time (due to the limit of available computing resources and logical sequence). After the model tuning, the production simulations for the manuscript were redone and the analysis and discussion of the results needed to be revised in the manuscript. We had therefore asked for an extension of three months for a thorough revision.

**C:** *Comment #1.2: The global burial fluxes of POC (110 PgC kyr-1) and calcite (115 PgC kyr-1) look good (see Table 1). Please check the highest POC data estimate in Tab. 1 (i.e., 2600 PgC kyr-1) is this not a settling flux?*

**R:** It is indeed the burial flux reported by Burdige (2007) and the high numbers refer to the continental margin sediments.

**C:** *Comment #1.3: There seems to be something wrong with the opal fluxes: How can the opal settling flux in areas ¿1km (79-84 Pmol Si kyr-1) be larger than the global flux (22- 40 Pmol Si kyr-1) in Table1? Also the units are different in the text (Tmol Si yr-1) and the Tab. 1 (Pmol Si kyr-1). And why is the global opal burial flux even larger (82 Pmol Si kyr-1). And compared to this the observed global burial is tiny (7.1 Pmol Si kyr-1).*

**R:** We thank the reviewer for pointing out the different units used in the table and text. In the revised version we only use $Pmol\,kyr^{-1}$ in Tab. 1 and the related text.

The opal burial in $R_{coupled}$ is $12\,Pmol\,kyr^{-1}$, not 82. We apologize for the typo. The same number (12.1) can be found again in Tab. 2.

The deep-water settling fluxes of opal in Tab. 1 are higher than the global ones, because the numbers stem from different studies and none of them reported both for the deep-water and global sediments, except for Treguer et al. (1995) who reported an opal burial in deep-water sediments of 5.9 and in the global sediments of $7.1\,Pmol\,kyr^{-1}$. This was discussed in L335–L340 in the submitted version

[Figure]

Figure 3: Temporal evolution of the atmospheric $pCO_2$ in the experiments with an addition of 1000 ($R_{pert1k}$) and 2000 PgC ($R_{pert2k}$) in the atmosphere (a); and the change of calcite content in sediments relative to the state before the $CO_2$ perturbation in $R_{pert1k}$ (b) and $R_{pert2k}$ (c).

and we rephrased it to make it clearer.

**C:** *Comment #2: Figure 6: The new color-bars for the model results (right) are finer than the color-bar for the observations (left). Therefore, model and observations cannot properly be compared as more features appear in the model results than in the observations (this was introduced to address one of the comments of reviewer #1).*

**R:** Thanks! This inconsistency was overlooked in the submitted manuscript. The plots are redone now with the same color bar.

**C:** *Comment #3: A transient experiment or paleo-application. This would be highly informative and was suggested by both reviewers (e.g., 1st comment of Reviewer #1) but has not been addressed. At least the model could be used to simulate an idealised perturbation experiment to showcase that it can be used for transient applications and that the sediment properties respond. In particular, because this is mentioned as one of the main motivations for configuring this model.*

**R:** Please also see the reply to the general comment above. To demonstrate how the model responds to perturbation in the carbon system, we added 1000 ($R_{pert1k}$) and 2000 PgC ($R_{pert2k}$) into the atmosphere at the end of the coupled simulation $R_{coupled}$ and let them run for another 2000 years. The interactions between the atmosphere, ocean and sediment carbon pools, particularly in the form of calcite, were examined. Calcite in sediments are strongly dissolved (Fig. 3b and c) and the atmospheric $CO_2$ declines from 1235 ppm to 490 ppm in $R_{pert2k}$ and from 765 ppm to 380 ppm in $R_{pert1k}$ after 2000 years (Fig. 3a), showing the effect of carbonate compensation feedback. New sections have been added in the method and result chapter in the revised manuscript to describe these new experiments.

**C:** *Comment #4: I was also not very convinced by the authors reasoning for why they did not include some output of C-isotopes. This was also suggested by both reviewers and is highly relevant for a model that will be applied to paleo-applications.*

**R:** We agree – with both reviewers – that having isotopes in a coupled ocean–carbon cycle–sediment is highly relevant for a model that is intended to be used for paleo-applications, the more since the sediment model allows for the construction of synthetic cores that could be directly used for model-data comparisons. However, contrary to what the comment suggests, there are not such results, there is no "output of C-isotopes" that we could possibly present. The coupling has been technically realised (the code exists), but has neither been tested, nor calibrated. First of all, the calibration

and validation of the carbon isotopic part of the model is obviously dependent on that of the underlying carbon cycle and it goes without saying that there is no point in starting to calibrate the isotopic part as long as the fundamental C cycle does not yield satisfactory results (we are convinced that the present revision is now bringing us closer to this milestone). It is furthermore well-known that an ocean-only model (without sediments) already takes an order of magnitude longer to equilibrate its C isotope distributions than it requires to equilibrate its DIC and ALK distributions (see, e.g., Lynch-Stieglitz et al. (1995)); adding ocean–sediment exchange processes further extends this time. Besides these timescale constraints, which are inherent to the system under study, it should not be forgotten that additional tracers will also lead to longer computation times: for the sediment part alone, adding $^{13}C$ and $^{14}C$ will increase the computation time by $\sim 180\%$ (computation time roughly scales with the square of the number of tracers considered). Adding robust C-isotope results in this paper would simply require such a large amount of time, alone for the computations. Practically speaking, requiring us to include isotope results at this stage raises an unrealistic, if not impossible expectation.

Please rest assured though that tackling the isotopic related parts in the coupled model is one of our first priorities once the fundamental carbon cycle is deemed to operate in a satisfactory way, the more since isotopes also offer further means to improve the calibration of the sedimentary process representations.

**C:** *Comment #5: Why a mass correction to account for the asynchronous coupling is necessary is unclear to me. Please explain and justify this better in the text. My understanding is, that while this might affect the inventories & fluxes during the spin-up phase the system should eventually come to a (new) steady- state.*

**R:** We thank the reviewer for insisting on this mass-conservation issue, which was actually addressed in response to a comment to the initial version of the manuscript, but unfortunately incompletely assessed in the previous revision. The reviewer's understanding about the asymptotic behaviour of the asymptotic error due to the asynchronous coupling is justified: this error should fade away when steady-state is approached (at least on average over a few coupling intervals)

For the current revision, we further examined the mass conservation in the different compartments of the coupled model. Mass conservation in MEDUSA2 is very strong (the relative error between the actually calculated global inventory change and that calculated from the globally integrated transport and reaction terms is typically $10^{-9}$–$10^{-12}$). Decreasing the coupling interval by a factor of 5 (from 50 to 10 years) only improves the inventory imbalance from $0.53\% \, \mathrm{kyr}^{-1}$ to $0.4\% \, \mathrm{kyr}^{-1}$ Although the shorter time step clearly contributes to reduce the diagnosed imbalance, the improvement is comparatively small. We therefore conclude that the asynchronous coupling cannot be the main reason for the imbalance. It is most likely due to the (known) tracer conservation issues related to the use of an unstructured grid in FESOM. This will be further investigated and improved in future development work of FESOM-REcoM. In the meantime the mass correction helps us to ensure that the total tracer inventories in the ocean is strictly conserved, considering the globally integrated in- and output fluxes of the different tracers.

The paragraph on the mass conservation issue and how it was addressed has been rewritten and moved to the section on FESOM2.1-REcoM3p in the model description chapter.

**C:** *Comment #6: The higher spatial resolution result (Rhigh): I understand, that this was included*

*to address parts of the 2nd comment of Reviewer #2. However, these results are rather out of place here. Also the authors argument that this experiment shows that the coarse model resolution of Rcoupled is mainly responsible for the low settling fluxes (lines 316–320) is at least insufficient: It is unclear if model resolution is the only boundary condition/parameter that changes between both configurations. Anyhow, there are multiple ways to increase the POC settling fluxes in the lower resolution Rcoupled configuration (e.g., by increasing the low export production as acknowledged by the authors, or decreasing pelagic POC remineralization rates).*

**R:** The resolution in this run was indeed not the only difference between those simulations. The ecosystem is a bit different in the high-resolution run, including a second zooplankton class which affects the cycling of nutrients. During the revision we conducted one simulation with the same code version but a much higher resolution (with ca. 127,000 surface grids) and updated the numbers for $R_{high}$ in Tab. 1. The fraction of shallower-water POC settling is about 70% of the global flux, supporting the argument that the low resolution of simulations in this study is mainly responsible for the underestimation of the fraction of the shallower-water POC flux. The reviewer is right that results of the high-resolution run in the table seems to be out of place, since it is not the focus of this study and not discussed anywhere else in the manuscript. However, it is an important evidence for the considerable role of model resolution which can be better illustrated when comparing with the low-resolution run in numbers. Thus, we kept it in the table but added a short explanation in the table caption.

**C:** *Comment #7: Fig. 3: I suggest including the profiles at the end of Rcoupled here as well because these are the new results presented and tested in the manuscript.*

**R:** Agreed. The profiles of $R_{coupled}$ are moved from the appendix to Sect. 3.2.5 'Impact of the complex sediment on productivity and nutrient supply' and those of $R_{sedbox}$ to the appendix.

**A few minor comments:**

**C:** *Fig. 6: What is the simulated wt% here? Is it mean over the bioturbated layer, at 10cm, or something else? And what are the corresponding values of Hayes et al. (2021) representing?*

**R:** Our model result was shown as the averaged sediment wt% over the surface 10 cm, while Hayes et al. (2021) shows the averaged wt% over the Holocene age if the age constraints are available and otherwise the measured compositions reported for the surface sediment. In the revised manuscript, for the comparability to data, we still calculated the averaged sediment wt% over the surface 10 cm, even when the reactive layer in our MEDUSA application is extended to 50 cm.

**C:** *Fig. 7: I find it not very intuitive why you show RNO3 here. Can you please motivate this. If sulfate reduction would be included in the model (more important than Mn, or Fe-reduction on a global scale) then one could nicely show the different fractions.*

**R:** Following the referee's implicit suggestion, we included sulfate reduction (skipping Mn(IV) and Fe(IV) reduction) as a further redox process in the revised version of the model. In the wake of this extension, Fig. 7 has been replaced with the carbon degradation rate by aerobic respiration, nitrate and sulfate reduction and the corresponding discussion in the manuscript has been updated.

**C:** *Fig. C1: Why is there such a large drop in the calcite burial flux even though the calcite settling flux increases over the experiment?*

**R:** In the submitted version there was no drop in calcite burial but in opal burial. Since the coupled simulation was rerun, all the related figures have been updated. In the new Fig. C1 the change in burial flux of opal is around 0% during the entire coupled simulation. Only POC burial shows a drop at the beginning of the simulation which can be explained by the drop of POM settling flux on top sediment.

We show this figure in the appendix to provide evidence that the ocean-sediment system starts to move towards a new steady state after a relative rapid adjustment during the first one or two coupling cycles. The small trend at the end of 2000 years indicates that a system including the complex representation of the seafloor sediments needs much longer to reach the steady state. The main focus here is the long-term trend, not the larger change during the adjustment. Thus, the figure does not resolve the first 200 years with more details. The large changes in the beginning of the simulation are caused by the large gradients at the water-sediment interface, since the starting conditions used here were obtained with REcoM3p run with the sedimentary input from the original one-box sediment layer and with no riverine input. Once the coupled simulation with the (pre-charged) sediment start, the in- and output flux patterns in the ocean completely change. Since the sediment now buries POC, less DIN is returned to ocean bottom, and less oxygen is consumed there. Also already during the first coupling cycle, biological productivity adjusts due to changes in nutrient supply. Accordingly the deposition, and thus degradation and diffusive return flux patterns adapt, and the whole system evolves towards a new steady-state, as shown on these graphs.

**References**

D. J. Clements, S. Yang, T. Weber, A. M. P. McDonnell, R. Kiko, L. Stemmann, and D. Bianchi. New estimate of organic carbon export from optical measurements reveals the role of particle size distribution and export horizon. Global Biogeochemical Cycles, 37(3):e2022GB007633, 2023. doi: 10.1029/2022GB007633.

John P. Dunne, Jorge L. Sarmiento, and Anand Gnanadesikan. A synthesis of global particle export from the surface ocean and cycling through the ocean interior and on the seafloor. Global Biogeochemical Cycles, 21(4):GB4006, 2007. doi: 10.1029/2006GB002907.

Christopher T. Hayes, Kassandra M. Costa, Robert F. Anderson, Eva Calvo, Zanna Chase, Ludmila L. Demina, Jean-Claude Dutay, Christopher R. German, Lars-Eric Heimbürger-Boavida, Samuel L. Jaccard, Allison Jacobel, Karen E. Kohfeld, Marina D. Kravchishina, Jörg Lippold, Figen Mekik, Lise Missiaen, Frank J. Pavia, Adina Paytan, Rut Pedrosa-Pamies, Mariia V. Petrova, Shaily Rahman, Laura F. Robinson, Matthieu Roy-Barman, Anna Sanchez-Vidal, Alan Shiller, Alessandro Tagliabue, Allyson C. Tessin, Marco Van Hulten, and Jing Zhang. Global ocean sediment composition and burial flux in the deep sea. Global Biogeochemical Cycles, 35 (4):e2020GB006769, April 2021. ISSN 0886-6236, 1944-9224. doi: 10.1029/2020GB006769.

C. Laufkötter, J. G John, C. A. Stock, and J. P. Dunne. Temperature and oxygen dependence of the remineralization of organic matter. Global Biogeochemical Cycles, 31(7):1038–1050, 2017. doi: 10.1002/2017GB005643.

Jean Lynch-Stieglitz, Thomas F. Stocker, Wallace S. Broecker, and Richard G. Fairbanks. The influence of air-sea exchange on the isotopic composition of oceanic carbon: Observations and modeling. Global Biogeochemical Cycles, 9(4):653–665, 1995. doi: 10.1029/95GB02574.